# RETRIEVAL AUGMENTED THOUGHT PROCESS FOR PRIVATE DATA HANDLING IN HEALTHCARE

## ABSTRACT

Large Language Models (LLMs) have demonstrated the strong potential to assist both clinicians and the general public with their extensive medical knowledge. However, their application in healthcare is constrained due to concerns about the privacy of data used in training, which prevents the integration of private and personal information because of security and ethical issues. Moreover, if their capabilities can be enhanced with information retrieval to access up-to-date knowledge, the current integration of LLMs with Information retrieval lacks robustness to imperfect retrieval, which can hinder their effectiveness and even reduce overall performance. In this work, we address this challenge by introducing the Retrieval-Augmented Thought Process (RATP). Given access to external knowledge, RATP formulates the thought generation of LLMs as a multiple-step decision process. To optimise such a thought process, RATP leverages Monte-Carlo Tree Search and learns a proxy reward function that permits cost-efficient inference. On a private dataset of electronic medical records, deliberately excluded from any LLM training set, RATP achieves 35% additional accuracy compared to in-context retrieval-augmented generation for the question-answering task.

## 1 INTRODUCTION

Recent advancements in Large Language Models (LLMs) trained on extensive datasets have showcased their enhanced capabilities in diverse tasks such as question-answering (Kamalloo et al., 2023), conversational abilities with humans (Bubeck et al., 2023) and notably providing medical knowledge (Singhal et al., 2022; Lee et al., 2023b). Yet, their usage in healthcare is hindered by their limited proficiency in accessing and accurately handling private data.

**Private data is any sensitive knowledge deliberately kept unavailable to LLMs during training due to ethical or business considerations**. This includes, for example, medical records (Pampari et al., 2018; Jia et al., 2020; Alsentzer et al., 2022) or banking details. These data must be excluded from large language model training sets because once learned, their privacy cannot be ensured (Kim et al., 2023; Zeng et al., 2024a; Carlini et al., 2023). This creates a significant barrier to LLM usage in healthcare as organisations cannot risk potential Personal Identifiable Information (PII) leaks, especially under regulations such as GDPR which protect closely the usage of such data. Furthermore, private databases are frequently updated, sometimes daily, making the knowledge embedded within a model's parameters quickly outdated. Continuously retraining the model with new data involves significant computational and financial resources (Brown et al., 2020) which can be afforded by very few organisations. Hence, relying solely on LLMs, regardless of their increasing size, is impractical.

Thus, in scenarios such as administrative tasks (e.g., letters, discharge summaries) (Thirunavukarasu et al., 2023) or decision aids (semi-autonomous diagnostic)(Lee et al., 2023a), where accessing information from a private database is required, the fusion of LLMs' capabilities with external knowledge sources becomes crucial. While Retrieval-Augmented Generation (RAG) (Lewis et al., 2021) has introduced the LLM and information retrieval pairing with the embedded cross-product between a query and the documents, recent works have shown its lack of robustness to unsuccessful retrieval (Yao et al., 2023c), to the extend that it can negatively impact performance (Yoran et al., 2024).

Additionally, beyond the privacy issue, external knowledge can potentially address some of the inherent limitations of LLMs (Delétang et al., 2023). Importantly, it enables the LLMs to recall

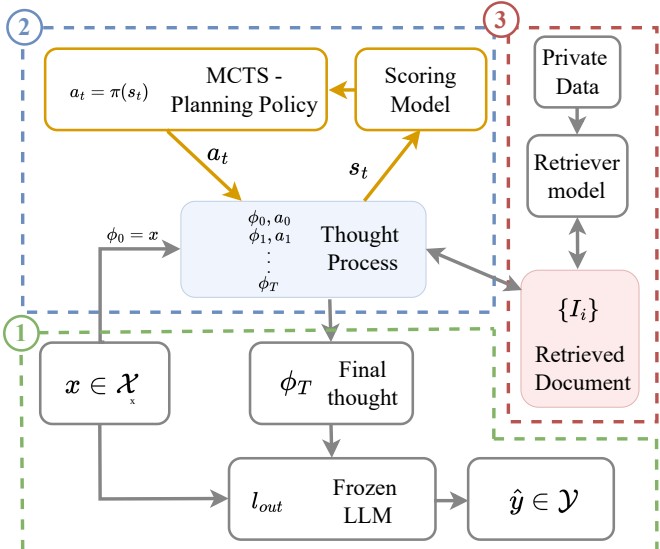

Figure 1: **Retrieval-Augmented Thought Process overview.** ① The frozen LLM $l_{thought}$ given an answer $\hat{y}$ to the question $x$ by using the extra context $s_T$. ② The thought process starts from the question $x$ and outputs the best thought found $s_t$ to help answering $x$. The actions $\{a_i\}$ are decided by the MCTS with feedback from the scoring model. This component is detailed in Figure 2. ③ The information retrieval system interacts with the thought process by answering its queries with retrieved documents $\{I_i\}$.

information from outside of their limited context window which is particularly beneficial in multi-turn dialogues (Xu et al., 2024). Furthermore, grounding LLMs with factual external knowledge reduces the production of factual inaccuracies and hallucinations (Borgeaud et al., 2022; Zhang et al., 2023), which are major issues impacting their performance, especially in less common domains or when dealing with private data. Moreover, using external knowledge allows the clinician to trace back the source of the information, unlike the implicit knowledge stored in model parameters. Hence, it has the potential to improve the accuracy and transparency of responses, addressing two noted limitations of LLMs that hinder their impact in healthcare (Thirunavukarasu et al., 2023). Consequently, this approach could support various medical applications detailed in Appendix A.

However, current literature still faces several desiderata to be fulfilled simultaneously for LLM and external knowledge pairing:

1. Guaranteeing data **privacy**.
2. Processing batches of documents beyond the **LLM's context window**.
3. Exploit **reasoning** capabilities of LLMs to filter out irrelevant or noisy information.
4. Ensuring **Transparency** of the retrieval-augmented thought process.

To fulfil these desiderata we introduce the Retrieval-Augmented Thought Process (RATP) in Figure 1. RATP enhances the thought generation capabilities of LLMs by treating it as a multi-step decision-making process utilising external knowledge sources.

**Our contributions** include:

1. Formally, we formulate the open-book question-answering task as a sequential decision-making problem. In light of this new formalism, we compare existing methods and, in collaboration with clinicians from different specialities and countries, highlight the significance of retrieval-augmented thought processes for healthcare applications.

2. Methodologically, we introduce the RATP, which leverages Monte-Carlo Tree Search to combine the reasoning capabilities of Large Language Models and the access to external knowledge.

3. Empirically, using publicly available LLMs, we evaluate RATP's effectiveness on the emrQA dataset and EHRQA dataset, two QA datasets on real medical materials (electronic medical records (EMRs) and discharge summaries) kept private from LLMs training.

## 2 PRELIMINARIES

**The Question Answering Task** We consider the general setting of question-answering where an autonomous machine learning model provides answer $\hat{y} \sim \mathcal{Y}$ given an input query $x \sim \mathcal{X}$. Given the state-of-the-art performance achieved by LLMs, in our work we consider the machine learning model to be a general-purpose LLM $\ell$. The loss of the task can be evaluated through a metric $R$ with $\mathbb{E}_{(x,y)\sim(\mathcal{X},\mathcal{Y})} R(\hat{y}, y)$, where $\hat{y} = \ell(x)$

**Open-Book QA** In healthcare applications, there are many scenarios where an external knowledge database is a must to answer questions as privacy concerns and/or computational costs prevent this knowledge from being encoded in the LLM's weight. For instance, redacting discharge summaries or pre-screenings of patients are tasks that require the information contained in private electronic medical records. Without loss of generality, we use $\mathcal{K}$ to denote the space of external knowledge, the LLM $\ell$ then answers the query with a subset of the knowledge, such that $\hat{y} = \ell(K, x)$ — as in the most general cases, the knowledge database can be too large to be fed to the $\ell$. The problem of finding the most appropriate piece of information is known as the information retrieval (IR) problem:

$$K^* = \arg\max_{K \in \mathcal{K}} R(\hat{y}, y) \quad \text{where} \quad \hat{y} = \ell(x, K). \tag{1}$$

## 3 RETRIEVAL-AUGMENTED THOUGHT PROCESS

In this section, we first use formal language to define the retrieval-augmented thought process as an MDP in Sec. 3.1; we then introduce Monte-Carlo Tree Search as an efficient and effective planning algorithm to solve the MDP in Sec. 3.2; finally, we discuss practical implementation choices of the scoring model in MCTS in Sec. 3.3.

### 3.1 MULTI-STEP THOUGHT GENERATION AS A MARKOV DECISION PROCESS

Formally, we define the multi-step thought generation as a **Markov Decision Process** (MDP), denoted as $\mathcal{M} = (\mathcal{S}, \mathcal{A}, P, r, \mathcal{X}, T)$. $\mathcal{S}$ is the state space. In our context, the state space is the space of *thought process* — the current reasoning graph leading to the answers. A state $s$ is composed of previous thoughts $\phi$ and previous actions. $\mathcal{A}$ is the action space; in our context, the action space contains all possible combinations of thoughts in the current thought process $s_t$. For ease of notation, any external document $K$ is also considered to be a thought. $P$ is the dynamics, in our context, the **transition dynamics is known** and instantiated as feeding the combined thoughts to a properly prompted LLM $\ell_{\text{thought}}$ to generate a new thought, i.e., $s_{t+1} = (s_t, a_t, \phi_{t+1})$ with $\phi_{t+1} = \ell_{\text{thought}}(a_t)$. $R$ is the reward function that evaluates the final answer. We denote the query $x$ as the initial *thought* $\phi_0 = x$ (hence the thought process

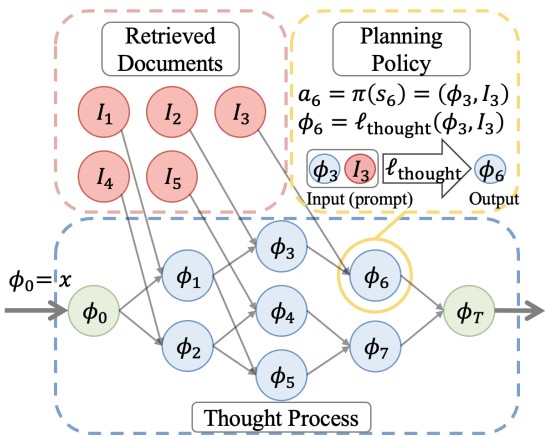

Figure 2: **Modeling the thought process.** Each thought is generated from previous thoughts and/or documents, effectively creating a graph. The planning policy controlling the construction of this graph is detailed in Figure 3.

also initializes with $s_0 = x$). Therefore, the query distribution $\mathcal{X}$ is the initial state distribution. $T$ is the problem horizon — the maximal number of reasoning steps.

Starting the query $s_0 = x$, an external **policy** $\pi : \mathcal{S} \mapsto \mathcal{A}$ determines whether the LLM continues to develop the existing thought with either external knowledge or another existing thought, i.e., $a_0 = \pi(s_0)$. Then, the LLM $\ell_{\text{thought}}$ transits the thought process (state) to $s_1 = (s_0, a_0, \phi_1)$ with $\phi_1 = \ell_{\text{thought}}(a_0)$, and an action $a_1$ will be generated by $a_1 = \pi(s_1)$, consequently, we have $s_2 = (s_1, a_1, \phi_2)$ with $\phi_2 = \ell_{\text{thought}}(a_1)$, $a_2 = \pi(s_2)$, ... Such a thought process terminates until the maximal reasoning timestep $T$ is reached, and the final thought $\phi_T = \ell_{\text{thought}}(a_{T-1})$ will be used to generate the final answer of the initial query: $\hat{y} = \ell_{\text{out}}(x, \phi_T)$, where $\ell_{\text{out}}$ denotes a frozen LLM. The reward function $R$ provides an episodic scalar feedback only at the end of a thought process:

$r_T = R(\hat{y}, y)$, and $r_{t<T} = 0$. In our work, we use $s_\pi = (\phi_0, a_0, \phi_1, a_1, ..., \phi_T)$ to denote the above thought process generated with policy $\pi$, and consider the policy optimization problem:

$$\pi^* = \arg\max_\pi \mathbb{E}_{(x,y)\sim(\mathcal{X},\mathcal{Y}), \hat{y}\sim s_\pi} [R(\hat{y}, y)]$$

## 3.2 PLANNING WITH MONTE CARLO TREE SEARCH

Given the vast action space in the above problem and the fact that the system dynamics model (i.e., the thought generation LLM $\ell_{\text{thought}}$) is accessible during inference, we choose to use Monte Carlo Tree Search (MCTS) (Coulom, 2007; Browne et al., 2012) for effective and efficient decision optimization. MCTS explores the tree of decisions and returns the best action found. In our case, we use MCTS to build the graph of thoughts and documents. Guided by our scoring system, MCTS realizes the trade-off between exploiting the best thought and exploring other thoughts. In the tree-search, the root $\phi_0$ is the initial query $x$ and each node is a thought $\phi$. The children of a given thought are the thoughts generated from it. The 4 key steps of MCTS are **Selection**, **Expansion**, **Simulation**, and **Backpropagation**. Figure 3 illustrates one step of the MCTS decision process by summarising the action of these four functions. This step is repeated until the answer is found or we reach the thought process decision step limit $T$.

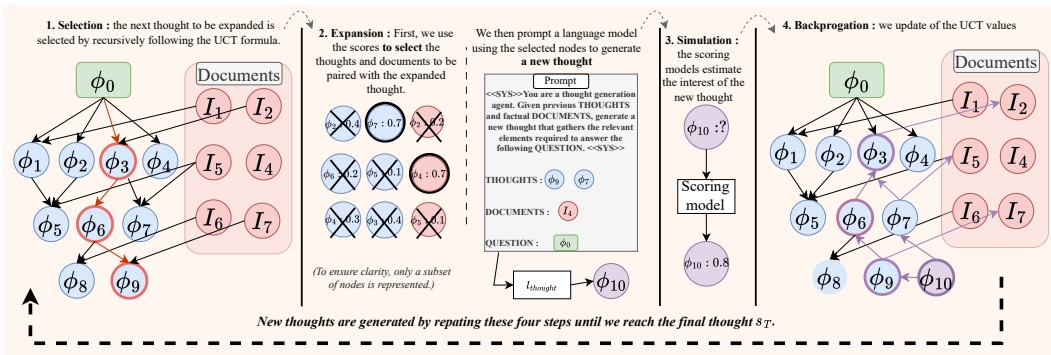

Figure 3: **One complete step from our MCTS decision process.** It is divided into four functions, which are repeated until we find the answer or the thought process size limit is reached. The **Selection**, **Expansion**, **Simulation**, and **Backpropagation** functions are described in section 3.2. Their associated algorithm can be found in Appendix G.

**Selection.** In the selection, we decide which thought to be expanded. From the root $\phi_0$ of our graph, we explore the graph until finding a thought $\phi_i$ that has never been used to generate another thought: a node without children. The exploration is guided by the Upper Confidence bounds applied to Trees (UCT) formula (Kocsis & Szepesvári, 2006; Sabharwal et al., 2012). When we meet a node with children, we choose to explore the child with the highest UCT value. For the $i^{th}$ thought that has been visited $n_i$ times, with a score $q_i$, and whose parents have been visited $N_i$ times, the UCT value expression with an exploration parameter $c$ is $\frac{q_i}{n_i} + c\sqrt{\frac{ln(N_i)}{n_i}}$

**Expansion.** The thought $\phi_i$ selected in the previous step is paired with other thoughts (including external documents) to generate a new thought. The choice to choose documents or thoughts for the pairing is decided by the probability $p_{\text{doc}}$ and the scoring model (see section 3.3). Then, the chosen thoughts and documents are combined with a prompt template (see examples in Appendix B.1) and given to $l_{\text{thought}}$ to generate the new thought.

**Simulation.** In our case, the simulation function computes the score of the new thought by using one of the scoring models described in section 3.3.

**Backpropagation.** This function recursively updates the UCT value of every node that has led to the new thought. Starting from this new node, it uses the new thought's score and the updated visit counts in the UCT formula to modify the UCT value of each parent until the root or documents are reached.

**Documents retrieval.** As the number of documents in the collection can be very large, we perform an initial document retrieval step using traditional retriever models Additional retrieval queries are made when needed. The initial implementation of the framework used Contriever (Izacard et al.,

2022a) but other retrieval models, in particular for different modalities (images, tables), can be used to broaden the applicability of the RATP framework.

### 3.3 Scoring models

**Oracle Score** To score the value of each thought, a naive way is to use the QA LLM $\ell_{\text{out}}$, which takes the query $x$ and any given thought $\phi$ as inputs and outputs the answer to the question. An oracle score can be defined by comparing this generated answer to the true answer with the metric score $R$ of the QA task. However, such a perfect signal is **by definition not available at inference time**. Therefore, two scoring models are introduced as proxies of such signals. Details on the implementation and comparison of the scoring models can be found in Appendix C.

**Model-Based Estimation.** In the first approach, we propose a model-based method to learn a proxy reward function $R_\theta$ that estimates the oracle scores using a training dataset. Specifically, when running MCTS on the training data, we would be able to generate an offline dataset that contains various thought combinations $(\phi^{(i)}, \phi^{(j)})$, and whether those thoughts are informative enough to solve the original query $x$:

$$r^{(i,j)} = R(\ell_{\text{out}}(\ell_{\text{thought}}(\phi^{(i)}, \phi^{(j)}), x), y).$$

Given such an offline dataset $\mathcal{D} = \{\phi^{(i)}, \phi^{(j)}, r^{(i,j)}\}_{(i,j)}$, $R_\theta$ can be optimized through

$$\theta \leftarrow \arg\min_\theta ||R_\theta(\phi^{(i)}, \phi^{(j)}) - r^{(i,j)}||^2 \tag{2}$$

and used as a proxy of the Oracle score during inference time when we do not have access to the true answers. $R_\theta$ is typically instantiated as an MLP or XGboost model.

**Self-critics score.** However, acknowledging that such an offline dataset may not always exist. We also introduce the second approach based on LLM self-critic. In the literature, using an LLM as a self-criticism agent is widely applied to reflect on a previous LLM answer and improve it (Welleck et al., 2022; Chen et al., 2023; Gou et al., 2023). It has been reported that LLM can detect their own mistake when prompted with their previous outputs. Thus, we leverage this ability by asking the LLM to predict if a thought is accurate and contains all the information to answer the query.

## 4 Related Work

Table 1: **Comparison with Related work.** The desiderata laid down in Section 1 are considered. RATP is the only method that fulfills them all.

| Method | Guarantee Privacy | LLM-Training-Free | Unconstrained context | Reasoning Ability | Interpretable |
|---|---|---|---|---|---|
| Pre-trained LLM | ✗ | ✗ | ✗ | ✗ | ✗ |
| Fine-tuned LLM | ✗ | ✗ | ✗ | ✗ | ✗ |
| RAG | ✓ | ✓ | ✗ | ✗ | ✗ |
| Self-RAG | ✗ | ✗ | ✓ | ✓ | ✓ |
| RATP | ✓ | ✓ | ✓ | ✓ | ✓ |

**Private data and LLM** Recent studies have extensively demonstrated the vulnerability of private data processed during LLM training (Plant et al., 2022; Kandpal et al., 2022; Panda et al., 2024; Carlini et al., 2023; Zeng et al., 2024a; Kim et al., 2023). The primary issue is the LLM's memorisation of data, which cannot be prevented during pre-training (Carlini et al., 2023) or fine-tuning (Zeng et al., 2024a). This memorisation makes LLMs susceptible to malicious extraction attacks. Since no method fully prevents this memorisation (Kandpal et al., 2022), the safest approach is to exclude private data from training datasets and use RAG instead (Zeng et al., 2024b). Despite recognition of the problem, few solutions have been proposed. To the best of our knowledge, this is the first paper to propose and benchmark a solution using a private dataset.

**Thought Processes as New Prompting Strategies** Recent advancements in prompting strategies have greatly improved the reasoning capabilities of Large Language Models. This progress began with the *Chain of Thought* (CoT) approach by (Wei et al., 2023), which breaks down complex problems into simpler steps, known as *thoughts*. Building on CoT, subsequent research has focused

Table 2: **Theoretical comparison between RATP and other thought processes** with the lens of the Information Retrieval as Multi-step decision-making problem formalism.

| Thought Process | MDP Characterization[*] | Learning Target[†] | MDP Solver |
|---|---|---|---|
| LLM | $T = 0, \mathcal{A} = \emptyset$ | $\pi : \mathcal{X} \mapsto \mathcal{Y}$ | Pre-Training |
| RAG | $T = 1, \mathcal{A} = \mathcal{K}$ | $\pi : \mathcal{X} \mapsto \mathcal{K}$ | $\min_k \in \mathcal{K} \|k - x\|^2$ |
| CoT | $T = 1, \mathcal{A} = \hat{\mathcal{X}}$ | $\pi : \mathcal{X} \mapsto \hat{\mathcal{X}}$ | $\min_{\hat{x}} \in \hat{\mathcal{X}} \|\hat{x} - x\|^2$ |
| ToT | $\mathcal{A} = [N]$ | $\pi : \mathcal{X} \mapsto \mathcal{A}^T$ | BFS/DFS |
| GoT | $\mathcal{A} = \text{GraphOPs}$ | $\pi : \mathcal{X} \mapsto \mathcal{A}^T$ | Graph Pattern |
| Self-RAG | $\mathcal{A} = \mathcal{K}^*$ | $\pi : \mathcal{X} \mapsto \mathcal{K}^*$ | LLM Fine-Tuning |
| RAT | $T = N, \mathcal{A} = \mathcal{K}$ | $\pi : \mathcal{X} \mapsto \mathcal{X}$ | $\min_k \in \mathcal{K} \|k - x\|^2$ |
| RATT | $\mathcal{A} = \mathcal{K}^N$ | $\pi : \mathcal{X}^{\mathcal{N}} \mapsto \mathcal{X}$ | $\min_k \in \mathcal{K} \|k - x\|^2$ |
| HippoRAG | $\mathcal{A} = \mathcal{K}$ | $\pi : \mathcal{K} \mapsto \mathcal{K}^2$ | Personalized PageRank |
| RATP | $\mathcal{A} = \{\mathcal{K} \cup [N]\}^d$ | $\pi : \mathcal{S} \mapsto \mathcal{A}$ | MCTS |

[*] RAG: the action is selecting a document; CoT: the action is selecting a prompt from a set of expert demonstration $\hat{\mathcal{X}}$; ToT: the action is selecting from the $N$ generated thoughts; GoT: the action is defined as different graph operators; Self-RAG: the action is selecting the relevant segments from the retrieved document; RAT: the action is selecting a document for each thought; RATT: the action is selecting a subset of the documents; HippoRAG the action is selecting a document; RATP: the action is selecting a subset of the external documents and existing thoughts of size $d$.

[†] RAG: the learning target is to prompt the query with external knowledge; CoT: the learning target is a prompting policy that selects the prompt for each given query; ToT (and GoT): the learning target is to find the reasoning path out of the generated tree (or graph) of thoughts; RAT: the learning target is modifying a thought; RATT: the learning target is summarising a set of thought; HippoRAG: the learning target is building the Knowledge Graph; RATP: the learning objective is to find the **markovian** optimal thought generation paths; therefore, it is **much easier than searching for the thought sequences directly** as in ToT and GoT.

on generating, combining, and selecting these thoughts. Notable methods include the *Tree of Thought* (ToT) (Yao et al., 2023b) and *Graph of Thought* (GoT) (Besta et al., 2023), which offer greater flexibility in thought combination. Similar to our approach, other strategies employ Reinforcement Learning techniques, such as Monte Carlo Tree Search, to explore the thought space in constrained reasoning problems like the Game of 24 and the 8-puzzle (Ding et al., 2023). However, these methods rely heavily on LLM-generated content and are susceptible to hallucination, limiting their use in real healthcare applications. Therefore, integrating these strategies with information retrieval is essential for grounding them in reality. Additional presentation and comparison of popular thought processes can be found in Appendix F.

**Pairing Information Retrieval and LLM** Integrating Information Retrieval with Large Language Models significantly enhances the efficiency of accessing and processing vast amounts of information (Shuster et al., 2022; Ai et al., 2023; Peng et al., 2023). One method involves pre-training LLMs with IR models like Contriever (Izacard et al., 2022a; Guu et al., 2020; Izacard et al., 2022b). Another method fine-tunes LLMs with task-specific tokens (Yao et al., 2023a) or adds layers to the model structure (Hu et al., 2023), though these approaches require significant computational resources and are vulnerable to extraction attacks. Alternatively, Retrieval-Augmented Generation (Lewis et al., 2021; Ram et al., 2023) leverages LLMs' in-context learning abilities to integrate IR data without extensive additional training, thus avoiding sensitive data leaks. Iterative prompting techniques like RAT or RATT (Wang et al., 2024; Zhang et al., 2024; Feng et al., 2023; Yu et al., 2023) enhance their reasoning capabilities by integrating external knowledge from documents in a chain-of-thought manner and/or with LLM fine-tuning (Asai et al., 2023), addressing the vulnerability of LLMs to misleading documents (Ren et al., 2023). For the same reason, HipporRAG (Gutiérrez et al., 2024) uses an LLM preprocessing of the document collection. Table 2 illustrates that the MDP formalism developed in section 3.1 can analyse and compare all the popular iterative prompting strategies.

Our approach uniquely enhances RAG by safely and accurately handling healthcare-sensitive data, utilising frozen LLMs to prevent training data leaks. We are the first to propose a method that grounds a complete thought process, planned via reinforcement learning solutions, in factual documents for private information retrieval.

## 4.1 EXPERIMENTAL SETUP

**Main experimental setup: private medical datasets** We analysed and benchmarked our method on two real, private, and sensitive healthcare data. First, we consider unstructured EMRs. These

EMRs have been gathered from 2004 to 2014 at the Partners HealthCare System, Boston, during the i2b2 project (examples in Appendix B.1). They are paired with the emrQA dataset (Pampari et al., 2018), which consists of open-ended medical questions based on patient records. These questions were created in coordination with physicians and medical experts. To evaluate our performance on open-ended questions, we used the Exact Match metric from SQuAD (Rajpurkar et al., 2016; Izacard et al., 2022b; Ram et al., 2023). (Details in Appendix B.1.) Second, we focus on discharge summaries. The EHRQA dataset (Bardhan et al., 2022) contains multiple-choice questions derived from patients' discharge summaries from the MIMIC-IV (Johnson et al., 2020) database, which includes anonymized patient records from Beth Israel Deaconess Medical Center between 2008 and 2019. The questions are categorized into eight types: Treatment, Assessment, Problem, Etiology, Sign/Symptom, Vitals, Test Results, History, Instruction, and Plan. Each question has been reviewed and refined by three clinicians. A crucial property of both these datasets is that their access is controlled preventing its use for LLM training.

**Additionnal experimental setup: Open-Domain Question Answering.** We add a second experimental setup to highlight the difference between the private and public knowledge settings. For the latter, we consider the open-domain question-answering task using the Boolq dataset (Clark et al., 2019). This widely-used dataset consists of closed-ended questions based on Wikipedia common knowledge. (The analysis of this setting is in Appendix D).

### 4.2 ANALYSIS

**Baselines.** In this subsection, our methods are compared against two prevalent zero-shot open-domain question-answering approaches. The first baseline, **LLM** ($\hat{y} = \ell_{\text{out}}(x)$), involves directly prompting an LLM with the question. The second baseline, **RAG** ($\hat{y} = \ell_{\text{out}}(x, k)$), represents the in-context IR method (Lewis et al., 2021; Ram et al., 2023). Here, the LLM generation process is augmented with one document $k$ retrieved from the knowledge base by Contriever (Izacard et al., 2022a).

**Oracles** methods use the oracle scoring system that requires the answers to the questions. As such, these methods do not apply to answering new questions. They are included to demonstrate the importance of information retrieval and the potential of our methods with an ideal scoring model. **MCTS oracle** is the full method with the oracle score, and **MCTS oracle w/o IR** is identical but with the information retrieval function deactivated. The former can generate thoughts from documents and previous thoughts, while the latter can only access previous thoughts ($\mathcal{K} = \varnothing$).

**RATP.** Two versions are introduced : **MCTS self-critic** and **MCTS estimation** are our MCTS algorithms using the self-critic and offline model-based estimation scores, respectively. **Tree-of-Thought with IR (ToT \w IR)** is an enhanced version of Tree-of-Thought (Yao et al., 2023b) which integrates Information Retrieval as delineated in Figure 2.

Table 3: **Ablation study for the Private Knowledge Question Answering task (emrQA dataset).** We report the Exact Match accuracy (%). The margins of error are computed by bootstrapped student-t tests (95%). The thought process size is 25.

| Baseline | | Oracle | | RATP | | |
|---|---|---|---|---|---|---|
| LLM | RAG | MCTS oracle w\o IR | MCTS oracle | ToT \w IR | MCTS self-critic | MCTS Estimation |
| 34 ($\pm$0.6) | 24 ($\pm$0.4) | 52 ($\pm$0.4) | 88 ($\pm$0.3) | 64 ($\pm$0.7) | 67 ($\pm$0.5) | 71 ($\pm$0.5) |

Even without access to patient records, the LLM correctly answers 34% of the time by leveraging its prior knowledge of standard prescriptions. For instance, when asked, *"What is the drug strength of aspirin prescribed to the patient?"*, the LLM correctly infers the default dosage of *1mg*. Table 3 indicate that adding a document to the prompt (RAG) leads to worse performance. This aligns with previous studies showing that RAG can be unreliable, depending on the dataset, retriever model, and knowledge base (Ram et al., 2023; Yoran et al., 2024). We identify two main reasons for this performance decline: current retriever models often fail to find relevant documents in large knowledge bases (Yao et al., 2023c), and LLMs can be confused by retrieved information, particularly when it is irrelevant or noisy (Yoran et al., 2024). ***Takeaway: Our experiment confirms that adding information in context can reduce LLM performance.***

The enhanced performance of our methods, underscores that our thought process can guide an LLM from an initially incorrect answer to a correct one : $\mathbb{E}_{(\mathcal{X}, \mathcal{Y})}[R(\ell_{\text{out}}(s_0), y)] \geq$

$\mathbb{E}_{(\mathcal{X},\mathcal{Y})}[R(\ell_{\text{out}}(s_T), y)]$. This phenomenon is evidenced by the increase in accuracy correlating with the size of the thought process, as depicted in Figure 4: the LLM's performance improves after processing multiple documents and generating several thoughts. Notably, the thought process not only allows the viewing of more documents compared to other methods but also strengthens resilience against both unsuccessful information retrieval and misleading documents, as exemplified in Figure 16 in the appendix. In this example, our MCTS method is not misled by irrelevant document 1 and 3 and eventually succeeds in extracting pertinent information from note 2. *Takeaway: the knowledge provided by RATP increases substantially the number of correct answers.*

Even with the oracle score, RATP is not flawless. Incorrect answers occur when the LLM cannot revise its stance within the thought process size limit, $T$. This happens if all retrieved documents are irrelevant, the LLM cannot use them effectively, or it fixates on an incorrect answer. The limit $T$ is set to prevent excessively long inferences. As shown in Figure 4, the number of LLM queries increases linearly with the number of thoughts, leading to significant financial or time costs for extensive thought processes. Typically, our methods find the correct answer within 20 thoughts, as indicated by the accuracy plateau in Figure 4. In contrast, the oracle score accuracy rises to 88% with 25 thoughts. *Takeaway: LLMs may need an excessive number of thought generation steps to pivot, exceeding an acceptable cost.*

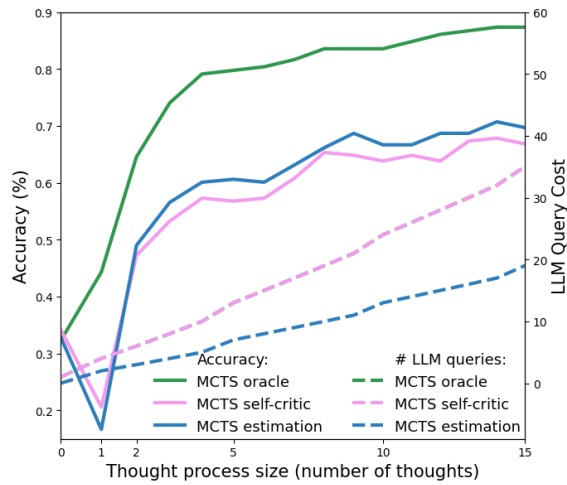

Figure 4: **Evolution of the accuracy and the number of LLM queries on the emrQA dataset**. When we increase the thought process size (i.e. the number of thoughts generated), the accuracy increases but the number of LLM queries too.

**Ablation study:** Comparing the MCTS oracle with and without IR shows that access to retrieved documents accounts for a 37% increase in final performance. Surprisingly, a similar experiment in an Open-Domain public data setting (see Appendix D) also demonstrates that IR boosts performance. This result is unexpected, as the LLM has been extensively trained on Wikipedia articles, suggesting little benefit from retrieving documents from this source. However, this can be explained by the imperfect memory of LLMs (Delétang et al., 2023) and their propensity for hallucination (Zhang et al., 2023). It is widely recognised that prompting LLMs with relevant factual documents helps reduce hallucinations (Yu et al., 2023; Gao et al., 2024), thereby enhancing performance. In Appendix E, we present an additional experiment demonstrating the impact of IR on improving the general thought quality. *Takeaway: IR improves performance, even when the LLM has been trained on the knowledge base.*

Moreover, we use the ToT with IR method to highlight the performance gains brought by the MCTS algorithm. Unlike standard thought processes, which solely follow the scoring model's guidance, MCTS optimizes an exploration-exploitation trade-off modelled by the UCT equation. This allows it to develop poorly scored thoughts as well, leading to two interesting properties: a) Robustness against scoring system flaws, as a thought mistakenly flagged as irrelevant has a non-zero probability of being developed. b) It fosters more "creativity" by allowing concurrent and/or connected lines of thought. *Takeaway: the MCTS algorithm demonstrates superior performance compared to other thought process algorithms.*

Table 3 demonstrates the effectiveness of the offline model-based estimator. It matches the performance of the self-critic score while being significantly faster, requiring fewer LLM queries compared to its self-critic equivalent (see Figure 4).

**High transparency.** Finally, we wish to highlight the high transparency of RATP. As demonstrated in the inference example in Figure 16, for each answer provided by our system, the entire thought process $s_\pi = (\phi_0, a_0, \phi_1, ..., \phi_T)$ is accessible. The final thought $\phi_T$ provides the rationale behind the generated answer, and additionally, we can trace the source of the information through the retrieved documents.

## 4.3 BENCHMARK

**Self-RAG**(Asai et al., 2023). This method relies on teaching the model to use a new ¡retrieval¿ token and a self-reflection thought process to assess the relevancy of the retrieved documents. Self-RAG resorts to LLM fine-tuning but can be evaluated on documents outside of its training corpus.

**RATT**(Zhang et al., 2024). This method generates multiple thoughts from the initial query, retrieves a document for each thought, and combines all documents and thoughts into a prompt to generate the final thought. This process iterates multiple times.

Table 4: **Benchmark of LLM and Information Retrieval pairing methods on both Public and Private dataset**. We report the binary accuracy (%) for the BoolQA dataset and the Exact Match accuracy (%) for the emrQA dataset and EhrQA datasaet. The margins of error are computed by bootstrapped student-t tests (95%). Additional details on the implementation of each method are given in Appendix B. This experiment has been conducted with Mixtral8x7B deployed locally.

| Dataset | LLM | RAG | Self-RAG | RATT | RATP |
|---|---|---|---|---|---|
| Private (emrQA) | 34 ($\pm$0.6) | 24 ($\pm$0.4) | 35 ($\pm$0.5) | 28 ($\pm$ 0.9) | **71** ($\pm$0.5) |
| Private (EhrQA) | 48 ($\pm$ 0.7) | 56 ($\pm$ 0.7) | 48 ($\pm$ 0.7) | 56 ($\pm$ 1.4) | **60** ($\pm$ 0.7) |
| Public (BoolQA) | 66 ($\pm$0.8) | 67 ($\pm$0.6) | 67 ($\pm$0.7) | 71 ($\pm$1.1) | **72** ($\pm$0.8) |

**Difference between the private and public knowledge setting :** This benchmark highlights the unique aspects of the private knowledge setting, which is rarely studied. As expected, the LLM baseline performance is much lower for questions requiring knowledge it hasn't been trained on, emphasising the importance of IR in the private setting. Additionally, the LLM shows greater confusion with RAG in the private setting, indicated by a larger drop in accuracy. We interpret this as a greater reliance on retrieved documents in the private setting, where the LLM cannot rely on its prior knowledge. This increased confusion may also be due to out-of-distribution formatting, such as the technical jargon, abbreviations, and layouts of EMRs and discharge summaries, which differ significantly from the text the LLM has been trained on. Similarly, Self-RAG performs poorly because the documents to be retrieved and processed differ greatly from its fine-tuning dataset. While Self-RAG realises multiple thought processes of depth 1, RATP performs one deep thought process which provides additional context and enables queries on information that might not be explicitly stated in the initial question. In RATT, queries to retrieve documents are derived from the same initial thought, leading to poor diversity and fewer different documents. Additionally, While RATP's scoring model filters the noise, RATT's aggregation method retains it, corrupting the thought process. Finally, we observe that RATP's final accuracy is similar in both settings, suggesting that performance depends more on the reasoning abilities of the LLMs than on the knowledge setting, as shown by a comparison of different LLM sizes in Appendix I. This benchmark validates the effectiveness of our approach on both public and private data. **For the Question Answering task on Electronic Medical Records, RATP a 35% improvement in accuracy**.

**Limitations.** One significant limitation of our work is the difficulty in obtaining publicly available large-scale datasets that are guaranteed not to have been part of the LLMs' training set. This increasingly challenging condition poses a problem for machine learning researchers aiming to conduct research on LLM in Healthcare. Furthermore, while the performance of our two scoring systems is similar, each has distinct limitations. The self-critic score is computationally intensive, effectively doubling the number of LLM calls required (see Figure 4). In contrast, the estimation score is more computationally efficient but requires a training dataset composed of questions and samples from the knowledge base. We also want to clarify that our focus is on one specific privacy concern: sensitive data leakage from the training set. Other threats, such as interception of retrieved data (Zeng et al., 2024b), profiling individuals (Staab et al., 2024), or contextual privacy (Mireshghallah et al., 2023), are beyond the scope of this paper. Finally, while our method may appear complex with several components, it is actually easy to apply and generalise to new datasets. We provide a guide for practitioners in Appendix H, demonstrating that the method only requires two steps and minimal tuning to be implemented

## 5 CONCLUSION

In this work, we introduced **RATP**, a new framework that pairs LLMs with private information using a Markovian multi-step decision process. Our method meets the four criteria outlined in the

introduction for such approaches. **1.** It does not rely on LLM training, ensuring that patient-sensitive data will not be leaked. **2.** By dividing the retrieval into a multi-step process, it can handle knowledge well beyond its context window size. **3.** It leverages MCTS to enhance the self-reflection and self-critique abilities of LLMs across many documents, making it robust to unsuccessful or noisy retrievals. This allows it to be twice as performant as other methods in navigating real EMRs. **4.** The entire process occurs in natural language, enabling each step to be verified by the practitioner.

RATP enables the safe integration of new sensitive private knowledge into LLMs. This inclusion of private data, previously excluded due to ethical or safety reasons, facilitates personalised applications of LLMs tailored to each patient's medical context. Moreover, its transparent reasoning across multiple documents of diverse natures and origins supports multifactorial decision-making, which is crucial for complex medical scenarios: additional healthcare applications enabled by RATP can be found in Appendix A. Furthermore, by eliminating the need for extensive training, our approach reduces barriers to LLM utilisation, making them more accessible for various scenarios, especially for organisations, institutions, or countries with limited resources.

**Future work:** Applying the RATP framework directly to other data modalities, such as images or tables, using multimodal LLMs and alternative retriever models would broaden the method's applicability. Table 3 highlights the importance of guiding thought generation with a scoring model trained on medical data. Thus, developing robust scoring models using larger fine-tuned LLMs, while preserving data privacy, is a significant research direction. Moreover, clinicians might still be the best scoring model, making a "human in the loop" approach particularly valuable. This method, being transparent and providing reasoning steps in natural language, is well-suited for such an approach.

**Ethics Statement:** Given the medical context, the use of LLMs must be carefully regulated and validated to avoid errors and ensure patient safety. To address this, we implemented two safeguards: (1) A multidisciplinary team of clinicians from various specialties and countries supervised the work to ensure adherence to healthcare best practices. (2) In Appendix J, we identified and examined potential biases in the data and LLM output across different subgroups, with no significant biases detected.

**Reproducibility Statement:** The methodology is detailed in Section 3.2. Code and instructions for accessing datasets (emrQA, EHRQA, and BoolQ) are provided in the supplementary materials. These datasets are publicly available for research purposes. Additional experimental details, including prompts and implementations of baselines and models, are presented in Appendices B and C. A guide to practitioners is included in Appendix H to help apply the method to new datasets.

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

# Appendix

## Table of Contents

# A  LARGE LANGUAGE APPLICATIONS IN HEALTHCARE ENABLED BY RATP

Table 5: **Examples of current healthcare issues that could be addressed using large language model applications with access to private and sensitive data.** RATP enables such applications.

| Healthcare Problem | Usecase for LLM | Required Private Data | Reference |
|---|---|---|---|
| The administrative burden of documentation, e.g. discharge letters, requires significant physician time (Becker et al., 2010)) | LLMs can rapidly summarize information on a patient's history and treatment | EHR data, patient name, date of birth, address etc. | (Thirunavukarasu et al., 2023; Clusmann et al., 2023; Becker et al., 2010) |
| Language and knowledge barriers between clinicians and patients are difficult to overcome due to limited time or skills | LLMs can translate medical documents into different languages or from medical terminology to plain language | Medical records, discharge letters, and other documents, which often contain personal patient information | (Zaretsky et al., 2024; Thirunavukarasu et al., 2023) |
| To discuss a challenging case, clinicians often require a partner. Colleagues with necessary expertise are rarely unavailable ad hoc | Using the encoded medical knowledge, access to medical research material, and patient data, LLMs can iteratively and on-demand engage in informed discussions with clinicians and augment medical reasoning | EHR data including but not limited to laboratory results, reports, personal characteristics of the patient (e.g., age, sex,...), medication etc. | (Lee et al., 2023a) |
| Pre-screening of patients for clinical trials | LLMs are able to extract patient data, such as demographic information, comorbidities, and treatments, to determine if they are eligible to be included in a clinical trial on trial inclusion/criteria. | Medical health records, demographic data, comorbidities, clinical trial selection criteria and protocols | (Idnay et al., 2021) |
| Discovering and exploring clinical phenotypes | Phenotyping patients with postpartum haemorrhage (PPH) using discharge notes from electronic health records Identifying these granular concepts accurately allows the development of interpretable, complex phenotypes and subtypes. | Clinical notes, discharge summaries | (Alsentzer et al., 2023) |
| Efficient triage through patient profile summarisation | These findings suggest that LLMs could accurately identify higher-acuity patient presentation using data extracted from patients' first Emergency Department documentation. | Clinical documentation, medical records | (Williams et al., 2024) |
| Clinical prediction | Predicting hospital length of stay, in-hospital mortality, and hospital readmissions | Unstructured Clinical Notes (site-specific data) | (Jiang et al., 2023) |

# B  EXPERIMENTAL DETAILS

For all our experiments, the large language model used is Mixtral8x7B (Jiang et al., 2024) locally deployed on 4 A100 GPU accelerators with 80GB VRAM. The batch of document retrieval from the knowledge base is performed by a Contriever model (Izacard et al., 2022a) in its standard configuration. This model has been pre-trained on CC-net and English Wikipedia.

## B.1 PRIVATE KNOWLEDGE: UNSTRUCTURED ELECTRONIC MEDICAL RECORDS

To realise our experiment on private knowledge, we use the question with fine-grained answers from the emrQA (Pampari et al., 2018) dataset. These are questions of the type "What is the dosage of —*medication*—?". We only keep the questions whose answers are of the form "X mg" with X a number. The 4436 questions filtered are split into a training set (3110 questions) and a test set (1000 questions). The knowledge base for this dataset is composed of unstructured electronic medical reports which are gathered in patient records. For each patient, we re-split this patient record (Figure 5 provides examples) into 100-word chunks. During the retrieval step, the patient associated with the query is identified and the retrieved documents are these 100-word passages from its medical record. For this experiment, we directly embed the initial question as the input for Contriever.

**Discharge Diagnosis:**
Anterior left rib fractures [**2-6**]
Left pneumothorax
Bilateral pulmonary contusions
Left zygomatic arch fracture
Left lateral pterygoid plate fracture
Left lateral orbital wall fracture
Left minimally displaced orbital floor fracture
Bilateral lateral wall maxillary sinus fractures
Left ear skin avulsion

**Physical Exam:** VS: 95.6 90 138/77 18 98
GA: alert and oriented x 2, no acute distress
CVS: normal S1, S2, no murmurs
Resp: mild bibasilar crackles
[**Last Name (un) **]: soft, nontender, nondistended
Ext: warm, no edema, well perfused

**Pertinent Results:**
[**2113-1-22**] 01:23PM GLUCOSE-107* LACTATE-2.5* NA+-143 K+-4.6 CL–95* TCO2-31*
[**2113-1-22**] 01:21PM PT-13.1 PTT-24.1 INR(PT)-1.1
[**2113-1-22**] 01:21PM PLT COUNT-201
[**2113-1-22**] 01:21PM WBC-9.1 RBC-4.44* HGB-15.3 HCT-43.1 MCV-97 MCH-34.7* MCHC-35.8* RDW-13.1
[**2113-1-22**] 01:21PM ASA-NEG ETHANOL-NEG ACETMNPHN-NEG bnzodzpn-NEG barbitrt-NEG tricyclic-NEG
[**2113-1-22**] 01:22PM LIPASE-74*
[**2113-1-22**] 01:22PM UREA N-17 CREAT-1.1
[**2113-1-26**] 03:00AM BLOOD WBC-7.4 RBC-2.60* Hgb-9.0* Hct-24.7* MCV-95 MCH-34.8* MCHC-36.7* RDW-14.8 Plt Ct-135*
[**2113-1-26**] 03:00AM BLOOD Plt Ct-135*
[**2113-1-25**] 03:10AM BLOOD Glucose-84 UreaN-11 Creat-0.8 Na-137 K-3.0 Cl-102 HCO3-29 AnGap-10
[**2113-1-25**] 03:10AM BLOOD Calcium-8.1* Phos-2.6* Mg-2.1
[**2113-1-22**] CT SINUS/MANDIBLE/MAXIL: 1. Numerous facial fractures as enumerated before corresponding most closely to a Le Fort type III on the right and anterior,lateral and superior walls of the maxillary sinus on the left including the inferior orbital rim. 2. Hemorrhage contained within the maxillary sinuses. 3. Chronic non-displaced fracture of the tip of the dens.
[**2113-1-21**] CHEST (PORTABLE AP): Small right apical pneumothorax at the level of the second posterior interspace, slightly smaller than on [**3-26**]. Small right pleural effusion has probably developed. Lungs low in volume but clear. Moderate cardiomegaly unchanged. Healed right lower and lateral rib fractures.

**Service:** SURGERY
**Allergies:** No Known Allergies / Adverse Drug Reactions
**Chief Complaint**: left hemopneumothorax, bilateral pulmonary contusions, right rib fractures, multiple facial fractures, degloving injury to ear
**Major Surgical or Invasive Procedure:** left chest tube placement
**History of Present Illness:** 60M was struck by his car after attempting to do some mechanical work on it. Car rolled over him and dragged him several feet. He was brought to the ED by ambulance. He had concerns of RUQ pain, chest pain and a laceration to the head. He denied loss of consciousness. A chest tube was placed in the ED for hemopneumothorax. Following the chest tube placement, the patient had an episode of hematemesis and hypotension, but this resolved spontaneously. At baseline the patient is functional at home and take no anticoagulant medications. He suffered multiple facial fractures, an ear laceration, bilateral pulmonary contusions, multiple right sided rib fractures and a dens fracture.

Figure 5: **Examples of Unstructured Electronic Medical Records.** For privacy reasons, we present simulated EMRs resembling the actual dataset.

The prompts used to build the thought process in this experiment are the following.

The MCTS is configured with the following hyperparameters: exploration rate: $\sqrt{2}$, pick document probability: 1, thought sample size: 5, maximum thought process size: 25, score threshold to stop the thought process : (oracle: 0.5, self-critic: 0.49, model-based estimation: 0.9), retrieved document batch size: 5.

```
Using the given CONTEXT,
    answer
the following QUESTION. The
    unit
is in mg. Only output a
    number.

Exemples of answer: 30/900/10

CONTEXT : "{context}"

QUESTION : "{query}"

OUTPUT :
```

Figure 6: This prompt template is employed with the final thought to obtain the answer to the question. This is the same prompt template used in the RAG baseline.

```
Answer the following QUESTION.

The unit is in mg.
Only output a number.

QUESTION : "{query}"

OUTPUT :
```

Figure 7: This prompt template is employed to obtain the answer to the question when no additional context is provided. This is the prompt used in the LLM baseline.

Table 6: **Complete results for the Self-RAG method in both settings.**

| Method | Public dataset (BoolQ) | Private dataset (emrQA) |
|---|---|---|
| **Self-RAG (long)** | 67 | 20 |
| **Self-RAG (short)** | 57 | 35 |

## B.2 PUBLIC KNOWLEDGE : BOOLQ DATASET

The experiments on public knowledge have been realised by running our methods on the validation split from the Hugginface Boolq dataset. The knowledge base includes all the articles from the English Wikipedia. To create this database, we filter the most recent Wikipedia dump (December 2023) to only keep the body of the articles. Then, we merge and split these articles to create 500-word chunks of text. Thus, a retrieved document in this experiment is one of this 500-word passage. Moreover, the embedded query used as input to Contriever to retrieve a new batch of documents is the LLM's output from the "query" prompt template with the best-scored thought (see Figure 10).

The MCTS is configured with the following hyperparameters: exploration rate: $\sqrt{2}$, pick document probability: 0.5, thought sample size: 5, maximum thought process size: 10, score threshold to stop the thought process : (oracle: 0.5, self-critic: 0.49, model-based estimation: 0.21), retrieved document batch size: 2.

## B.3 ADDITIONAL METHODS

**Self-RAG:** the Self-RAG implementation relies on the official Self-RAG codebase. The largest model offered by the authors (Llama-13B) is being used on both our public and private dataset. The two settings proposed by the self-RAG authors (short-form and long-form text generation) have been tested with the default parameters. Only the best results are reported in Section 4.3. Full results are presented in table 6.

**ToT w\ IR:** We use standard hyperparameters that match the size of our thought processes (25) for ToT: step limit: 6, breadth limit: 2, size limit: 2. $P_{doc}$ that controls the integration of information retrieval is set to 1 for EmrQA and 0.7 for BoolQ.

```
As a thought generation agent,

your task is to analyze
    previous
THOUGHTS and a PATIENT RECORD.

From this information,
    generate
a new thought that compiles
relevant elements needed to
answer the following QUESTION.

Focus on identifying a
    quantity,
expressed in milligrams (mg),
    as
it appears in the PATIENT
    RECORD
or the THOUGHTS, before
considering the dosage.

THOUGHTS : "{thoughts}"

PATIENT RECORD : "{documents
    }"

QUESTION : "{query}"

RESPONSE :
```

Figure 8: This prompt template is employed to generate new thoughts for the amrQA experiment.

```
You are an agent that rates
    the
information contained in
    CONTEXT.
If the information contains
    in
the CONTEXT is accurate and
    you
have all the information
    required
to answer the QUESTION, you
output 1. If the CONTEXT is
    not
accurate or you don't have
    all
the information required to
answer the QUESTION, you
    output 0.

QUESTION : "{query}"

CONTEXT : "{thought}"

OUTPUT NUMBER :
```

Figure 9: This prompt template is employed by the self-critic scoring model.

## C  SCORING MODELS

### C.1  SCORING MODEL IMPLEMENTATION

**Training of the offline model-based score.** For both datasets the method to build a dataset and train the score estimation model is similar. We collect complete thought processes by answering questions from the training split of the dataset with our MCTS-oracle method. Thus, we are able to link every thought $\phi_t$ collected with its parents i.e. the thoughts or documents that have been used to generate $\phi_t$. The parents of $\phi_t$ are indicated by the action $a_{t-1} = (\phi^{t,1}, \phi^{t,2})$ because $\phi_t = \ell_{\text{thought}}(a_{t-1})$. As the thoughts processes have been generated by the MCTS-oracle policy, we also have the true score $R(\ell_{\text{out}}(\phi_t), x_i), y_i) = R(\ell_{\text{out}}(\ell_{\text{thought}}(\phi^{t,1}, \phi^{t,2}), x_i), y_i) = r^{(t,1,t,2)}$. Hence, we effectively have a dataset $\mathcal{D} = \{\phi^{(i)}, \phi^{(j)}, r^{(i,j)}\}_{(i,j)}$.

In practice, in the dataset $\mathcal{D}$, the feature vectors are the concatenations of the embedded versions of thoughts $\phi^{(i)}$ and $\phi^{(j)}$. These embeddings are generated by Contriever (Izacard et al., 2022a). The size of the dataset $\mathcal{D}$ is typically around 4000 samples. We trained gradient-boosting or MLP models on these datasets to perform the score estimations. The thresholds to compute the accuracy are chosen to maximize the precision on the training set. Finally, they are tested on the test split of their respective dataset.

**Self-critic scoring model.** The self-critic method entails prompting the LLM to reflect on its previous outputs. In our case, we combine the initial question and a given thought with a prompt template (see Figure 9) to query a frozen LLM about the accuracy and adequacy of the thought.

```
You are an agent that
    formulates
a document query. Given
    previous
THOUGHTS, formulate a query
    in
English to retrieve the
    relevant
information required to
    answer the
QUESTION.

THOUGHTS : "{thoughts}"

QUESTION : "{query}"

QUERY :
```

Figure 10: "This prompt template is employed to generate queries to retrieve documents with Contriever."

```
You are a thought-generation
agent. Given previous
    THOUGHTS
and factual DOCUMENTS,
    generate
a new thought that gathers
    the
relevant elements required to
answer the following QUESTION.

THOUGHTS : "{thoughts}"

DOCUMENTS : "{documents}"

QUESTION : "{query}"

RESPONSE :
```

Figure 11: This prompt template is employed to generate new thoughts for the Boolq experiment.

```
Using the given CONTEXT,
answer the following True
or False QUESTION. If the
answer is YES output 1. If
the answer is NO output 0.

CONTEXT : "{context}"

QUESTION : "{query}"

OUTPUT NUMBER :
```

Figure 12: This prompt template is employed with the final thought to obtain the answer to the question. This is the same prompt template used in the RAG baseline.

```
Answer to the QUESTION. If
the answer is YES output 1.
If the answer is NO output 0.

QUESTION: "{query}"

OUTPUT NUMBER :
```

Figure 13: This prompt template is employed to obtain the answer to the question when no additional context is provided. This is the prompt used in the LLM baseline.

In our framework, the LLM is constrained to respond with either 0(indicating the response is insufficient or inaccurate) or 1 (indicating adequacy and accuracy).To derive a more nuanced self-critic score, we examine the relative softmax probabilities assigned to the tokens "0" and "1" by the LLM. The self-critic score is calculated using the formula:

$$\text{Self-Critic Score} = \frac{\text{Softmax Score of Token 1}}{\text{Softmax Score of Token 1} + \text{Softmax Score of Token 0}}. \tag{3}$$

## C.2 Scoring Model Comparison

As detailed in section 3.3, we proposed two different scoring models to estimate the potential of the thoughts. These models should predict the oracle score which is not available at inference time. Thus, to assess the performance of these scores, we answer a subset of queries from the Boolq test set using our MCTS with the oracle score as the policy $\pi$. By doing so, we generate a set of pairs (question, thought-process) : $\{x_i, s_{\pi,i}\}$. For every generated thought of these thought processes, we compute

its score using the two scoring models. Finally, we compare how well these models predict the oracle score in Table 7.

In Table 7, it is evident that the model-based estimators outperform the self-critic model. The main reason for this disparity is that the estimation score model is specifically trained for this task while the self-critic score prompts a frozen LLM.

However, we also want to highlight that the weak performance of the self-critic model can be explained by its over-confidence. As depicted in Figure 14, the distribution of self-critic scores is more skewed towards 1 (indicating confidence that the thought contains the required information to answer the query) compared to the oracle score, which is skewed towards 0 (suggesting the thought is inaccurate or lacks information).

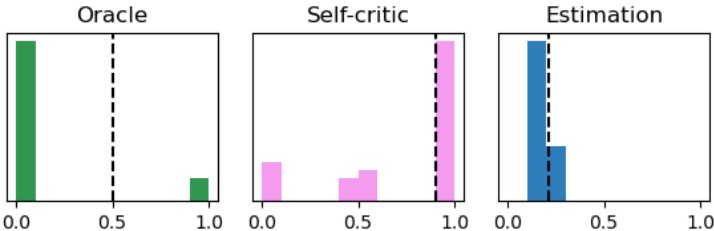

Figure 14: **Histograms of Score Distributions.** Each scoring system's score value is on the horizontal axis, and the vertical axis is the number of samples for the value. The dashed lines correspond to the thresholds used to compute the accuracy .

Table 7: **Comparison of Scoring Models.** These scoring models predict the oracle score for each thought from multiple runs of our MCTS method. The queries are from the Boolq test set, and we retrieved documents from Wikipedia. Accuracy is computed using thresholds computed on the training set (0.9 for self-critic, 0.21 for the model-based estimation).

| Models | Self-critics | Estimation |
|---|---|---|
| MSE | 0.60 | 0.12 |
| Accuracy | 42% | 73% |

The unbalanced distribution of oracle scores is a consequence of our MCTS implementation: we stop the generation of the thought process $s_\pi = (\phi_0, a_0, \phi_1, a_1, ...\phi_T)$ when the oracle score indicates that a thought $\phi_T$ has been generated which successfully answers the query. Therefore, in each run, only the last thought $\phi_T$ receives a high score, while preceding thoughts $\phi_0, \phi_1, ..., \phi_{T-1}$ that led to this final thought – either partially answering the query or being irrelevant – receive low scores.

However, if using LLMs as self-critics causes hallucinations in evaluating thoughts, the self-critic performance should improve with higher quality LLMs (more parameters). To verify how the quality of the LLM impacts the self-critic score's tendency to hallucinate and affects RATP's performance, we experimented with GPT-3.5-turbo and GPT-4, which are stronger general-purpose LLMs known to suffer less from hallucinations. This experiment was limited to the BoolQA dataset, as feeding sensitive data from emrQA to a non-locally deployed LLM raises ethical issues.

Table 8 supports our conclusion that the performance of RATP improves with stronger LLMs and fewer hallucinations in acting as self-critics. In addition, if LLMs that are better aligned with humans yield better results, it shows that leveraging human feedback would be an interesting way of improving the method.

## D   ADDITIONAL ABLATION STUDY : PUBLIC KNOWLEDGE SETTING

As seen in Table 9, IR still enhances performance even though the LLM has been trained on the knowledge base. There are two main differences compared to the private knowledge setting. First, the limited performance gain from RAG can be attributed to the nature of the retrieved documents,

Table 8: **Comparison of RATP (MCTS self-critic) performance using different LLMs as critic models.** The thought generation LLM is consistently Llama 70B.

| Self-critic Model | Accuracy |
|---|---|
| Llama-2 70B | 70 |
| GPT3.5-turbo | 73 |
| GPT4 | 81 |
| Oracle | 83 |

Table 9: **Ablation study for the Open Domain Question Answering task.** This comparison was conducted on the Boolq test set. The knowledge base was a Wikipedia dump from 2023. We report the binary accuracy. The margins of error are computed by bootstrapped student-t tests (95%). This experiment has been conducted with Llama-2 70B.

| Baseline | | Oracle | | RATP | | |
|---|---|---|---|---|---|---|
| LLM | RAG | MCTS oracle w\o IR | MCTS oracle | MCTS self-critic | ToT \w IR | MCTS Estimation |
| 64 ($\pm$0.3) | 65 ($\pm$0.2) | 80 ($\pm$0.5) | 83 ($\pm$0.4) | 70 ($\pm$0.3) | 68 ($\pm$0.7) | 70 ($\pm$0.3) |

which are Wikipedia articles. Both the retriever model and the LLM have been trained on Wikipedia, leading to more successful retrievals and better handling of the documents by the LLM. Second, the performance gain from IR is much lower; the difference for the Oracle method with and without IR is only 3%, which is expected given the nature of the knowledge base.

## E   HALLUCINATION EXPERIMENT

To demonstrate the impact of IR (even for public knowledge base), an additional experiments has been conducted to explicitly quantify the extent of hallucination in the generated thoughts. In these experiments, we evaluate the quality of thoughts produced by the MCTS+oracle scoring method, both with and without the integration of IR. The assessment uses the BoolQA test split as the dataset.

To effectively measure hallucination, we establish three proxy scores: **A. The oracle score**, as defined in the paper, which pertains to the downstream QA task performance of a frozen LLM prompted with the generated thought. **B. F1 and ROUGE-L** similarity metrics, which compare the generated thoughts with the gold standard paragraphs associated with the queries in the BoolQA dataset.

Table 10: **Assessement of the thought quality with and without IR.** The frozen LLM used is Mixtral8x7B.

| Metric | MCTS Oracle w/ IR | MCTS Oracle w/o IR |
|---|---|---|
| F1 | 0.134 | 0.080 |
| Rouge-L | 0.082 | 0.052 |
| Oracle | 0.317 | 0.255 |

Table 10 shows that across all these measures, the inclusion of IR in our method demonstrates a marked improvement in the quality of thought generation. There is a clear indication that IR contributes to generating higher-quality thoughts with reduced instances of hallucination and enhanced accuracy.

## F   COMPARISON WITH EXISTING THOUGHTS PROCESSES AND BASELINES

In this section, we contrast different thought processes with their graphical models.

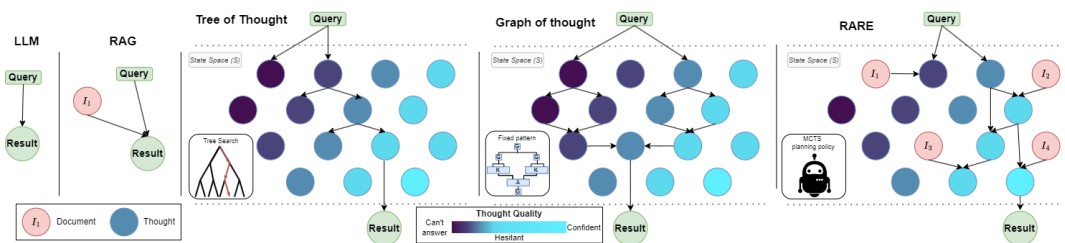

Figure 15: **Comparison with other popular types of thought process**

**Vanilla LLM for QA**    The naive approach is to directly query the LLMs for the answers. In this approach, there is no decision, and the answer is generated with a single-round dialogue with the LLMs.

**Retrieval Augmented Generation (RAG)**    In RAG, an external document can be used to alleviate the hallucination of the LLMs. The limitation of such an approach is the length of the document can be restricted by the context window size. In RAG, the LLMs answer the query within a single interaction. By formulating RAG as an MDP, the decision is 1-step, and the action can be defined as selecting the most relevant document form an external database.

**Tree of Thought (ToT)**    In ToT, the thought process is generated in a tree structure with specifically designed prompts that ask the LLMs to generate multiple thoughts at each timestep. To search over the generated tree-of-thought structure, either DFS or BFS can be used. Such a search is over the entire thought generation process, hence can be extremely high-dimensional and computationally challenging.

**Graph of Thought (GoT)**    In GoT, the thought process is generated as a directional graph. If we consider the thought generation process in GoT as an MDP, the action space is defined as the graph operators, such as aggregating or reflecting, and the exploration over such actions is conducted by LLMs.

**RATP**    in RATP, we solve the decision-making problem with an external planning policy learned with MCTS. Moreover, RATP has a more general State space where external documents are also considered to be *thoughts*. In addition, RATP can work with model-based return estimation such that in inference time the value of thoughts can be effectively estimated with light machine learning model — without using LLMs for self reflection.

## G    MONTE-CARLO TREE SEARCH IMPLEMENTATION

---

**Algorithm 1 Overview of the MCTS algorithm that builds our thought process.** The algorithms of the functions *Selection*, *Expansion*, *Simulation* and *Backpropagation* can be found in section G.

---

**Input:** query
**Initialization :**  documents,thoughts = [], [query]
**while** not thoughts[-1].is_terminal() **do**
    sThought = **Selection**(thoughts[0])
    nThought = **Expansion**(sThought)
    score = **Simulation**(nThought)
    **Backpropagation**(nThought, score)
    thoughts.append(nThought)
**end while**

---

---

**Algorithm 2** Selection

---

**Input:** thought
**if** thought.is_leaf() **then**
    Return thought
**else**
    **Selection**($\max_t$[t.UCT $|t \in$ thought.children])
**end if**

---

**Algorithm 3** Expansion

---

**Input:** sThought
**Initialization :** r = random float in [0,1[
**if** r > pDocument **then**
    **if** len(documents) = 0 **then**
        documents = **RetrieveDoc**(sThought)
    **end if**
    nThought = LLM.generate([sThought, documents.pop(0)]
    sThought.children.append(nThought)
**else**
    pThought = **SampleFromExistingThoughts**(1)
    nThought = LLM.generate([sThought, pThought])
    sThought.children.append(nThought)
    pThought.children.append(nThought)
**end if**
Return nThought

---

**Algorithm 4** Simulation

---

**Input:** nThought
score = **ScoringModel**(nThought)
Return score

---

**Algorithm 5** Backpropagation

---

**Input:** nThought, score
parents = nThought.parents
**for** p in parents **do**
    p.UCT.**update**(score)
    **Backpropagation**(p,score)
**end for**

---

## H   GUIDE FOR PRACTIONNER

In this section, we present a concise guide for implementing our method with new datasets.

**1. Choosing a Retriever Mode and LLM:** In our experiments, we use a pre-trained Contriever [4] with default parameters from its official GitHub repo. This model has been pre-trained on Wikipedia but can be used for any textual database as shown by our results on the electronic medical record. Moreover, as our method is built to be robust to unreliable retrievals, the choice of the retriever model should have a low impact. Similarly, while we provide the implementation for Llama-2, Gemma, and Mixtral, our method is directly compatible with any Large Language Models.

**2. Customizing Prompts:** Our methods use different prompt templates for the different steps (final answer, thought generation, self-critic). While we provide general-purpose prompts (see Appendix B.2), you can customize these prompts by incorporating expert knowledge (see Appendix B.1). This is particularly useful to enforce a desired shape for the output.

**3. (Optional) Training an Oracle Score Estimator:** The practitioner might want to use an estimator of the oracle score as a scoring model because it is cheaper in LLM queries with similar performance (see section 4.2). They would only need to provide a left-out dataset. The methods can then be run with the oracle scoring method and save every score thought. It will then train a model on such collected data points. We provide the training script for XGboost models.

**4. (Optional) Fine-Tuning MCTS Parameters:** To improve the performance of the method on a particular dataset, it is possible to adapt the values of some parameters.

- $P_{doc}$ : the probability of pairing a thought with a retrieved document or another previous thought.
- $c$ : the exploration parameter in the UCT formula, a higher value will favor the exploration of unseen thoughts, lower value favors the exploitation of highly scored thoughts.
- *Early stopping* : There are two mechanisms to stop the thought process. First a threshold, we stop the process when a thought has a score higher than a given value. Second, a finite size stops the thought process when a given amount of thoughts is reached.

# I GENERALISATION TO OTHER LARGE LANGUAGE MODELS

This section provides empirical evidence to support that RAPT can use any LLM models by performing a complete re-run of our experimental setup for Gemma 2B, Llama-2 70B and Mixtral8x7B.

Table 11 findings align with our expectations that larger models exhibit enhanced performance. Interestingly, the Gemma 2B model, being relatively smaller, does not exhibit significant gains from the RAPT method on the BoolQA dataset. This observation is attributed to the limitations of smaller language models in executing complex reasoning processes akin to self-reflection or chain-of-thought, which are critical for RATP's efficacy.

Table 11: **Comparison of RATP performance for different Large Language Models.** Unlike table 8, in this experiment the different LLMs are also used to generate the thought. We report the binary accuracy (%) for the BoolQA dataset and the Exact Match accuracy (%) for the emrQA dataset.

| Large language Model | Method | Private dataset (emrQA) | Public dataset (BoolQ) |
|---|---|---|---|
| **Gemma 2B** | Baseline LLM | 10 | 52 |
| | MCTS self-critic | 22 | 52 |
| | MCTS estimation | 38 | 40 |
| **Llama-2 70B** | Baseline LLM | 38 | 64 |
| | MCTS self-critic | 62 | 70 |
| | MCTS estimation | 65 | 70 |
| **Mixtral 8x7B** | Baseline LLM | 34 | 66 |
| | MCTS self-critic | 67 | 72 |
| | MCTS estimation | 71 | 65 |

# J ANALYSIS OF THE DATA BIAS

In this section, we analyse bias in the EMRQA dataset. Our method relies on LLMs, which have been demonstrated to have biases such as geographic bias (Manvi et al., 2024) or gender bias (Kotek et al., 2023). From the EmrQA case study, we identify three potential biases:

- **Geographic bias**: EMRs were collected at Partners HealthCare System in Boston.
- **Temporal bias**: EMRs were collected between 2004 and 2014.
- **Age bias**: The dataset excludes EMRs concerning minors.

Additionally, we provide further analysis of the data used regarding two criteria: age and sex (as the EMRs have been pre-processed to prevent identification, these are the only criteria we can reliably infer from the data). The results of these analyses are presented in Tables 12 and 13 13 of the supplementary PDF.

Table 12: **Accuracy of the method depending on the sex of the patient.** The sex of a subset of patients was manually retrieved from their EMRs. We report the accuracy (ExactMatch metric) of the answers to questions related to these patients.

| Sex | Proportion | Accuracy |
|---|---|---|
| Male | 60% | 69% ($\pm$ 2.6) |
| Female | 39% | 64% ($\pm$ 2.6) |

From this analysis, we conclude that the data used for the case study are slightly biased toward aged males. Some bias can be observed in the method's accuracy, as the method performs slightly better for younger patients and male patients.

Table 13: **Accuracy of the method depending on the age of the patient.** The age of a subset of patients was manually retrieved from their EMRs. We report the accuracy (ExactMatch metric) of the answers to questions related to these patients.

| Age | Proportion | Accuracy |
|---|---|---|
| **26-54** | 24% | 81% ($\pm$ 3.2) |
| **55-65** | 24% | 67% ($\pm$ 3.4) |
| **66-76** | 24% | 60% ($\pm$ 3.2) |
| **77-90** | 24% | 71% ($\pm$ 2.5) |

# K    EXAMPLE

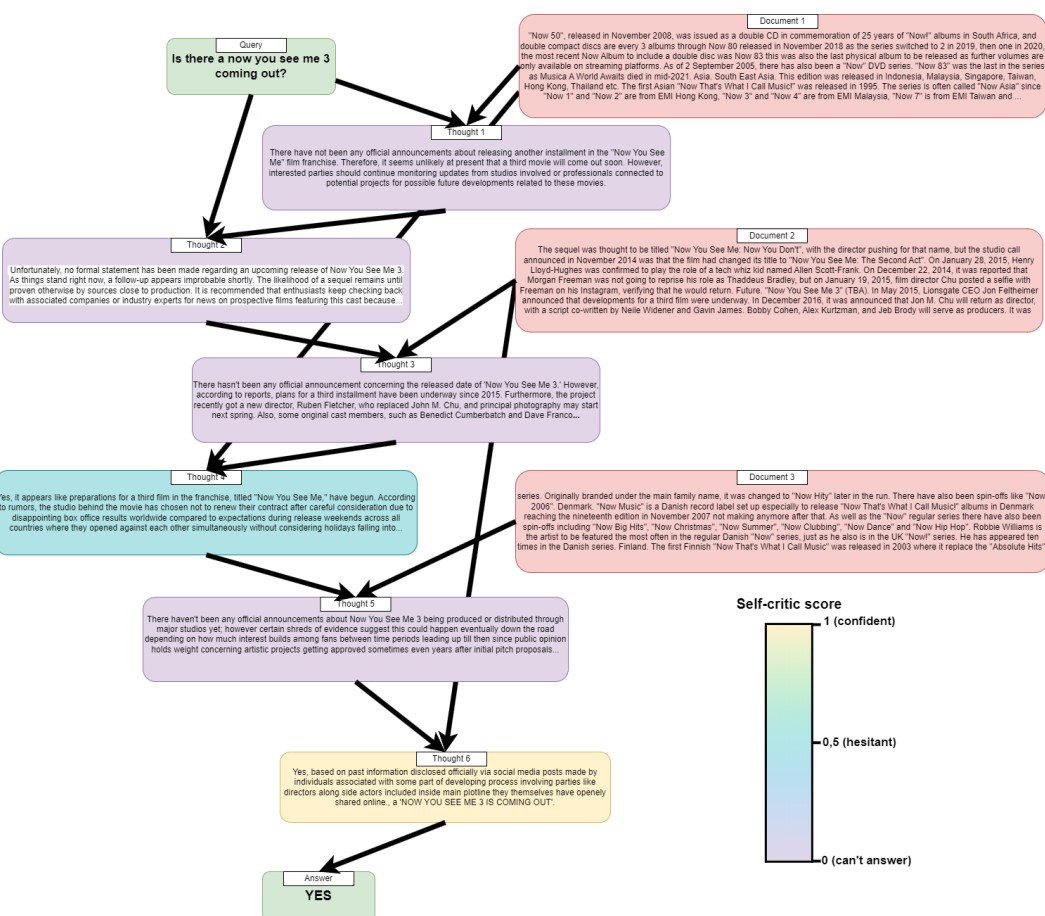

Figure 16: **Example of question answering by RATP.** It has been realized by the self-critic MCTS method on a question from the Boolq test set.

