# OpenReview forum: "Retrieval Augmented Thought Process for Private Data Handling in Healthcare"
_ICLR.cc/2025/Conference — Submitted to ICLR 2025_

### Official Review · Reviewer_vqsA · 2024-10-16

**Soundness:** 3
**Presentation:** 2
**Contribution:** 2
**Rating:** 6
**Confidence:** 3

**Summary:**

The paper describe a novel method to to expand LLM knowledge using recursive thought process that includes retrieving relevant documents. This method is extremely useful in the scenarios where private data is needed to answer a question.

**Strengths:**

According to the authors the method greatly increases the performance over the tested dataset.

**Weaknesses:**

- I think the paper is written in too abstract and over formulated manner, a concrete step-by-step walkthrough example of the process would be very useful to better understand  the process.

- What is the reason that RAG underperforms LLM in some cases, it is especially peculiar when we know the answers requires private data. Is the retrieval wrong? Does the model has issue utilizing the retrieved data? An analsys of the reason for failure would be useful and can contribute to the understanding of why your method is successful

- Figure 4 seems to be missing the MCTS oracle #LLM queries line, generally this figure is hard to read to the fact it has two y values, splitting the figure and adding the missing line would be useful

- For easier understanding of the method cost I think total tokens used would be easier to understand. Can total tokens per method be added to the analsys?

**Questions:**

None

---

> ### Author Response · Authors · 2024-11-25
> **Response to your review**
>
> We thank this reviewer for taking the time to review our work. We have carefully considered each point of feedback and provide our point-by-point responses below. Please don’t hesitate to let us know if any further clarifications are required.
>
> - (P1) Cost efficiency analysis.
> - (P2) Clarifying the presentation.
> - (P3) Failure of the RAG Baseline.
>
> ## **P1 Cost efficiency analysis.**
>
> We thank the reviewer's nice suggestion to provide an additional analysis regarding the inference cost of our method. Therefore we conducted an additional comparison of the token costs for each benchmarked method, which is provided in the global response. This new analysis improves the analysis against the baselines relevant to the reader and other reviewers.
>
> ## **P2  Clarifying the presentation**
>
> We thank the reviewer for highlighting the confusion in Figure 4. When ordered by equal MCTS size, the number of LLM calls for the oracle, and the LLM self-critic are identical, resulting in one line obscuring the other.
>
> In addition, we appreciate the reviewer's feedback regarding the clarity of the method's presentation. The multi-stage nature of our approach may contribute to this confusion. To address this, we point readers to Figure 3, which provides a step-by-step overview. Additionally, Appendix G includes detailed pseudocode for each stage of our Monte Carlo Tree Search algorithm instantiation, which will aid readers in understanding the method. We will ensure this pseudocode is noticeably referenced in the main text. Furthermore, we have included a concrete, step-by-step guide in Appendix H to assist practitioners in applying our method to new datasets or tasks. These elements will help clarify the methodology and resolve the reviewer's concerns.
>
> **Update:** Figure 4 has been updated to clearly distinguish the line representing the number of LLM queries for the MCTS oracle. Reference to the clarifying content in the Appendix has been highlighted in the main text.
>
> ## **P3  Failure of the RAG Baseline**
>
> To clarify the underwhelming performance of RAG on the private dataset, two main reasons were identified:
>
> 1. **Overperformance of the LLM Baseline**: General-purpose LLMs often provide default answers for medical questions even without additional context, which can align with correct treatments. For instance, when prompted with, *“What is the drug strength of aspirin prescribed to the patient?”*, the LLM correctly infers the default dosage of 1 mg. This inherent tendency improves their baseline performance.
>
> 2. **Confusion from Provided Context**: Supplying passages from patient EMRs can introduce irrelevant or noisy information, which confuses the LLM and leads to poorer answers. This aligns with prior findings showing that RAG's performance is dataset-dependent and can be negatively impacted by the retriever model or the knowledge base [1, 2]. Specifically, current retriever models often fail to identify relevant documents in large knowledge bases [3], and LLMs may struggle to handle retrieved information that lacks relevance or clarity [2].
>
> To address these issues, RATP incorporates two key strategies:
>
> - **MCTS Integration**: By using Monte Carlo Tree Search (MCTS) to integrate multiple retrieved documents based on dynamically generated queries, RATP increases the diversity of retrieved documents and improves the likelihood of accessing relevant information. The impact of the number of retrieved documents (scaling with the size of the MCTS) on task success is shown in Figure 4 of the manuscript.
> - **Scoring Model for Filtering**: A scoring model filters out irrelevant or noisy retrieved documents to mitigate confusion in the LLM. The positive correlation between RATP performance and the strength of the scoring model is demonstrated in Table 8 (Appendix D), with results summarized below:
>
> | **Self-critic Model** | **Accuracy (%)** |
> | --- | --- |
> | **Llama-2 70B** | 70 |
> | **GPT-3.5-turbo** | 73 |
> | **GPT-4** | 81 |
> | **Oracle** | 83 |
>
> These results highlight the importance of an effective scoring model in preventing RAG failures.
>
> **Update:** The reasons for RAG failure will be clarified in the revised manuscript.
>
> [1] In-context retrieval-augmented language models, 2023
>
> [2] Making retrieval-augmented language models robust to irrelevant context, 2024
>
> [3] ReAct: Synergizing reasoning and acting in language models, 2023
>
> ---
> We hope that we have sufficiently addressed the majority of the reviewers’ concerns and that this may encourage a reconsideration of their scores. We remain available and eager to engage in further discussions.

---

### Official Review · Reviewer_XAJF · 2024-10-28

**Soundness:** 2
**Presentation:** 2
**Contribution:** 2
**Rating:** 3
**Confidence:** 4

**Summary:**

This paper introduces a framework, Retrieval-Augmented Thought Process (RATP), designed to integrate external information retrieval with LLMs to safely and effectively use private medical data. It addresses challenges in privacy, reasoning, and robustness when LLMs access sensitive information for healthcare applications.

**Strengths:**

This paper take the privacy question in healthcare into consideration, which is very critical in the healthcare domain,  and propose corresponding  methods to enhances the interaction between LLMs and private data.

The experimental results comparing to proposed baselines looks promising in the EHR QA datasets.

**Weaknesses:**

1. Incremental Contribution:
While the paper addresses several critical issues in the healthcare domain—such as privacy, hallucination, and reasoning—it relies heavily on existing approaches, including RAG, multi-step thinking paths, and scoring systems, without introducing fundamentally new ideas.

2. Ambiguity in Baseline Models:
The authors mention using "LLM" as a baseline but fail to specify which LLM or group of LLMs they experimented with. This lack of clarity makes it difficult for readers to accurately assess the reproducibility and comparability of their results.

3. Lack of Comparison with Simpler Methods:
The paper does not compare the proposed method with simpler and widely adopted reasoning frameworks, such as Chain of Thought (CoT) with or without self-consistency. Including such comparisons would have provided better insights into the benefits of using the proposed Monte Carlo Tree Search (MCTS) framework.

3. High Computational Cost:
Although the use of MCTS improves the reasoning capacity of the LLM, the computational cost is likely much higher than that of simpler baselines. A detailed comparison of computational overhead, including the number of LLM queries and time or resource consumption, would strengthen the evaluation and help justify the trade-off between performance and efficiency.

4. Insufficient Coverage of Related Work:
The related work section lacks coverage of recent advancements in Medical LLMs and privacy in healthcare. Here are some works are notably missing:

Medical LLMs:
Li et al. (2023): ChatDoctor: A Medical Chat Model Fine-Tuned on a Large Language Model (LLaMA) Using Medical Domain Knowledge
Zhang et al. (2023): AlpaCare: Instruction-Tuned Large Language Models for Medical Applications
Kim et al. (2024): Health-LLM: Large Language Models for Health Prediction via Wearable Sensors

Health-domian privacy:
Zhang et al. (2022): Enhancing Small Medical Learners with Privacy-Preserving Contextual Prompting

Including these references would provide a more comprehensive understanding of related work in the healthcare domain, particularly concerning privacy and medical LLMs.

**Questions:**

1. Could you clarity what LLM indicates in the paper.

2. Could you compare with simple baselines such as COT with or without self-consistency.

3. Could you should the effectiveness of your model across basemodel family and size?

4. Could you show the computational cost comparison comparing to baselines?

---

> ### Author Response · Authors · 2024-11-25
> **Response to your review -- part 1**
>
> We thank this reviewer for their careful and detailed review of our work. Please find our answers as follows, along with corresponding updates to the revised paper:
>
> - (P1) Cost efficiency analysis.
> - (P2) Novelty concern.
> - (P3) Large Language Models used.
> - (P4) CoT as a baseline.
> - (P5) Related Work section.
>
> ## **P1 Cost efficiency analysis.**
>
> We appreciate the reviewer's encouragement to provide additional analysis regarding the inference cost of our method. As this point could concern other reviewers, an additional comparison of the token costs for each benchmarked method is detailed in the global response.
>
> ## **P2  Novelty concern.**
>
> We acknowledge the reviewer's point that our method builds upon existing works. Nonetheless, we wish to underscore the innovative aspects of our approach from three distinct angles:
>
> 1. **A New Formalism to Think About Thought Processes:** We propose a new way to understand and analyze thought processes by modeling them as sequential decision-making tasks within a Markov Decision Process (MDP) framework. This approach, as shown in Table 1 of our paper, enables the representation and comparison of various thought process methodologies like Chain-of-Thought [1], Tree-of-Thought [2], and Graph-of-Thought [3], etc. Furthermore, this framework integrates these thought processes into sequential decision-making, potentially opening opportunities for applying established MDP solutions, such as planning algorithms and reinforcement learning, to optimize iterative prompting strategies.
> 2. **Expanding from Structured Problems to Complex Healthcare Challenges**: Our approach builds upon previous methodologies through two major advancements. First, we uniquely ground thought generation in information retrieval, which reduces hallucination rates and enhances the ability to incorporate private external knowledge (e.g., EMRs) necessary for personalized question answering. Second, we adopt Monte Carlo Tree Search (MCTS), a more effective planning strategy compared to Breadth-First Search [2] or static graph patterns [3]. These innovations enable our method to transition from solving structured problems (e.g., the Game of 24, crossword puzzles) to addressing complex real-world challenges, such as answering personalized questions in healthcare while improving robustness to noisy or incomplete retrieval.
> 3. **Emphasizing the Integration of LLMs with Private Knowledge:** Our study addresses a key limitation of current methodologies: the inability of LLMs to utilize private, sensitive data like electronic medical records (EMRs), crucial in healthcare. RATP bridges this gap by enabling privacy-preserving access to external knowledge. On a private EMR dataset excluded from LLM training, RATP achieved a 70% improvement over the retrieval-augmented generation baseline in question-answering tasks (Table 3). This method enhances healthcare applications such as drafting discharge summaries, clinical decision-making, and patient phenotyping by improving response quality, interpretability, and trust through traceable reasoning.
>
> **Update:** We will clarify the merits and novelty of our methods in the introduction of our manuscript.
>
> [1 ] Chain-of-thought prompting elicits reasoning in large language models, 2023.
>
> [2] Tree of thoughts: Deliberate problem solving with large language models, 2023.
>
> [3] Graph of thoughts: Solving elaborate problems with large language models, 2023.
>
> ## **P3 Large Language Models used.**
>
> As described in the benchmark table's caption and the experimental details section of the appendix, the primary LLM used for the experiments, including the benchmark, is Mixtral-8x7b-Instruct [1].
>
> Additionally, in Appendix I, we investigate how RATP generalizes to other LLMs of varying sizes and families, specifically Gemma-2B and Llama-2-70B.
>
> **Update:** The revised manuscript will be updated to clarify this information more explicitly.
>
> [1] Mixtral of Experts, 2024

---

> > ### Author Response · Authors · 2024-11-25
> > **Response to your review -- part 2**
> >
> > ## **P4  “Chain-of-Thought” as a baseline.**
> >
> > Thank you for pointing this out. To clarify, if "Chain-of-Thought" (CoT) prompting refers to the technique of providing a series of expert demonstrations to guide the LLM before presenting a question (as described in [1]), we do not use this method in our approach or for any of the baseline models. While prompt engineering techniques such as CoT are widely applicable, fine-tuning each prompt for every dataset is infeasible given the large variety of possible configurations (5 benchmarked methods, each with different prompts).
> >
> > If "Chain-of-Thought" instead refers to a step-by-step reasoning process, with or without self-consistency, this is indeed a specific case of the Tree-of-Thought (ToT) approach. In our work, ToT with Information Retrieval is benchmarked as part of the ablation studies (see Tables 3 and 9). Specifically, CoT corresponds to a tree structure with a size and breadth limit of 1, which is included in the more general ToT framework. The ToT method has been shown to achieve a strictly superior performance compared to CoT [2].
> >
> > **Update:** We will add clarifications regarding the usage and comparison of "Chain-of-Thought" prompting in the related work section in the revised manuscript.
> >
> > [1] Chain-of-Thought Prompting Elicits Reasoning in Large Language Models, 2023
> >
> > [2] Tree of Thoughts: Deliberate Problem Solving with Large Language Models, 2023
> >
> > ## **P5  Related Work section.**
> >
> > We thank the reviewer for their suggestions regarding related works. While fine-tuned LLMs for healthcare are undoubtedly valuable, they address a research area that is orthogonal and complementary to our study. Our approach, RATP, can be combined with such models. However, these works, while potentially enhancing the performance of proposed baselines, do not focus on information retrieval (IR) or LLM pairing, unlike works that propose fine-tuning LLM to improve IR abilities such as Self-RAG [1], and are therefore not directly relevant as baselines.
> >
> > We agree that the work by X. Zhang et al. [2], which addresses contextual privacy, is relevant and should be included in the related work section. As noted in our limitations section, our study focuses explicitly on preventing sensitive data leakage from the training set and does not address contextual privacy issues. Additionally, while [2] provides valuable insights into privacy, it does not involve pairing IR systems with LLMs, making it unsuitable as a baseline for our study. However, future research could extend our framework to address broader privacy concerns by integrating methods like those in [2]. These could be particularly useful for mitigating contextual privacy risks in our framework's thought-generation and scoring model components.
> >
> > [1] Self-RAG: Learning to Retrieve, Generate, and Critique through Self-Reflection, 2023
> >
> > [2] Enhancing Small Medical Learners with Privacy-preserving Contextual Prompting, 2024
> >
> > **Update:** We will update the related work section in the revised manuscript to include a discussion on the different types of privacy concerns for LLMs, integrating the work of X. Zhang et al. [1].
> >
> > ---
> >
> > We hope that we have sufficiently addressed the majority of the reviewers’ concerns and that this may encourage a reconsideration of their scores. We remain available and eager to engage in further discussions.

---

### Official Review · Reviewer_7Jne · 2024-11-01

**Soundness:** 1
**Presentation:** 3
**Contribution:** 2
**Rating:** 3
**Confidence:** 4

**Summary:**

This paper presents the Retrieval-Augmented Thought Process (RATP), an approach designed to enhance large language models (LLMs) in healthcare by allowing them to interact with external knowledge sources while maintaining data privacy. RATP leverages Monte-Carlo Tree Search (MCTS) to support LLM decision-making in a multi-step reasoning framework, specifically addressing privacy concerns with private datasets excluded from LLM training. The approach yields substantial accuracy improvements over baseline retrieval-augmented generation (RAG) methods, notably achieving a 35% gain in accuracy on question-answering tasks over EMR datasets.

**Strengths:**

1. **Novelty**: The authors introduce RATP, which combines retrieval with a thought process through MCTS, enhancing LLM reasoning and robustness to imperfect retrievals.
2. **Focusing on Healthcare problem**: The RATP addresses privacy concerns which is highly relevant in the healthcare domain. By avoiding the use of sensitive data in training, the approach enhances the applicability of LLMs in handling sensitive healthcare information safely and efficiently.
3. Empirical Results: RATP demonstrates significant improvements in accuracy (35%) over standard retrieval-augmented generation methods on private datasets, underscoring its utility in healthcare QA task

**Weaknesses:**

1. **Privacy Concerns Still Existing**: While the training-free method mitigates some privacy risks by excluding sensitive data from training, it limits applicability to proprietary or on-premise LLM setups. Additionally, it raises questions about whether training-free alone is sufficient to address all privacy concerns, as data handling during inference still poses potential risks.
2. **Appropriateness of Experimental Setup**: The experimental setup for emrQA raises questions about the feasibility of accurately retrieving the "**oracle**" EMR. Since emrQA queries are designed to be answered directly from specific patient records, it is unclear how the system reliably identifies and retrieves the relevant EMR document (oracle) to support accurate answers. This critical detail is not fully explained in the paper, complicating understanding of the retrieval process and the model's effectiveness in real-world scenarios.
3. **High Inference Cost**: The RATP framework involves a multi-step thought generation process using Monte-Carlo Tree Search, which increases inference time and computational cost. Given the linear increase in LLM queries with the number of thought steps, extensive thought processes can become financially and temporally demanding, particularly in real-time applications.

**Questions:**

1. There appears to be an inconsistency in Figure 3 regarding document selection. In the "Expansion" step, there is no arrow indicating that \( I_4 \) is selected, yet it is included in the prompt for generating \( \phi_{10} \). This raises questions about how \( I_4 \) was chosen for the prompt without a connecting arrow in the illustration. Clarification on this selection process and whether \( I_4 \) should indeed be part of the prompt would improve the figure’s accuracy and help clarify the decision-making flow in the MCTS process.
2. As mentioned in the weaknesses, it would be beneficial to include a comparison of inference costs. Given the high computational demands of the multi-step thought process, a detailed analysis of inference costs compared to other methods would provide valuable insights into the efficiency of the RATP approach.

---

> ### Author Response · Authors · 2024-11-25
> **Response to your review**
>
> We thank this reviewer for taking the time to review our work. We have carefully considered each point of feedback and provide our point-by-point responses below.
>
> - (P1) Cost efficiency analysis
> - (P2) Privacy concerns
> - (P3) Retrieval implementation
> - (P4) Clarification
>
> ## **P1 Cost efficiency analysis**
>
> We thank the reviewer's nice suggestion to provide an additional analysis regarding the inference cost of our method. Therefore we conducted an additional comparison of the token costs for each benchmarked method, which is provided in the global response. This new analysis improves the analysis against the baselines relevant to the reader and other reviewers.
>
> ## **P2 Other Privacy Concerns**
>
> We understand the reviewer’s feedback regarding the privacy scope of our method As noted in the **Limitations** section of our manuscript, our primary focus is mitigating sensitive data leakage from the training set. While this is a critical privacy challenge that has hindered the deployment of LLMs in sensitive domains like healthcare, we acknowledge that other privacy threats, such as the interception of retrieved data, remain unresolved by our approach.
>
> The importance of addressing data leakage is emphasized in recent studies [1,2,3], which demonstrate how memorization of training data in LLMs can expose Personally Identifiable Information (PII). Our method offers a robust first step by eliminating this risk through the integration of external knowledge retrieval without retraining the model. Specifically, as detailed in Section 3.2, our retrieval-augmented thought process ensures that sensitive datasets like Electronic Medical Records remain outside the model's parameters while still improving question-answering quality.
>
> In future work, we plan to explore how RATP can be extended and integrated with complementary methods to address additional privacy challenges. For instance, contextual privacy risks could be mitigated by incorporating strategies such as privacy-preserving contextual prompting [4], which can enhance both the thought generation and scoring stages of our method. This direction aligns with the goal of developing a holistic framework for safeguarding sensitive data in healthcare.
>
> **Update:** We will expand the discussion in the limitations section to emphasize the potential for combining RATP with other methods to create comprehensive privacy safeguards.
>
> [1] ProPILE: Probing Privacy Leakage in Large Language Models, 2023
>
> [2] Exploring Memorization in Fine-tuned Language Models, 2024
>
> [3] Quantifying Memorization Across Neural Language Models, 2023
>
> [4] Enhancing Small Medical Learners with Privacy-preserving Contextual Prompting, 2024
>
> ## **P3 Retrieval Implementation**
>
> In our experimental setup for emrQA, each patient's electronic medical records (EMRs) are divided into 100-word chunks, forming a corpus uniquely identified by the Patient ID. The corpus size ranges from 2 to 98 records, averaging 29 records per patient. Each question in the dataset refers to a specific Patient ID, enabling retrieval of the corresponding corpus. Retrieval within a patient's EMR corpus is performed using a standard information retrieval model, specifically Contriever.
>
> **Update:** Appendix B.1 will be updated in the revised paper to clarify these details in the experimental section.
>
> ## **P4 Clarification of Figure 3**
>
> We thank the reviewer for identifying the typo in Figure 3. In the expansion step, documents were incorrectly labeled as $\phi_i$ instead of $I_i$. Additionally, in the illustration, $I_4$ is correctly selected due to its high score during the selection process. A backpropagation arrow from $\phi_{10}$ to $I_4$ should also be included.
>
> **Update:** The typo in Figure 3 has been corrected.
>
> ---
>
> We hope that we have sufficiently addressed the majority of the reviewers’ concerns and that this may encourage a reconsideration of their scores. We remain available and eager to engage in further discussions.

---

> > ### Comment · Reviewer_7Jne · 2024-11-28
> >
> > The reviewer have thoroughly search the details about the authors clarification. However, the reviewers' concerns are not fully handled through the rebuttal. Thus, I will remain my score.

---

### Official Review · Reviewer_SFUb · 2024-11-04

**Soundness:** 2
**Presentation:** 1
**Contribution:** 2
**Rating:** 3
**Confidence:** 4

**Summary:**

The authors addressed privacy data leakage in language model training and inference. They proposed a novel retrieval-augmented thought process that formulates multi-step thought generation as a Markov Decision Process, effectively solving it using Monte Carlo Tree Search. They evaluated the proposed method and baselines on two biomedical and two open-domain QA tasks, demonstrating the effectiveness of their approach.

**Strengths:**

+ The authors propose a novel approach that integrates RAG and MCTS into a single framework.

**Weaknesses:**

+ The experimental results are not sufficient to demonstrate the effectiveness of the proposed method.
For instance, S. Soni and K. Roberts [1] showed that they achieved an EM accuracy of 70.56 on emrQA using the training dataset and ClinicalBERT.
Since one of the proposed reward models requires the training data, the authors should include additional baselines that utilize the training data of each dataset.
Furthermore, X. Zhang et al. [2] proposed a pipeline that improves SLM performance by incorporating medical contexts from LLMs in privacy-restricted scenarios.
Including such baselines would strengthen the authors' contribution.

+ The authors do not conduct any privacy analysis.
Although they used a frozen LLM, the reward model was trained on a privacy-sensitive dataset, which methods like the model inversion attack [3] could extract training data.
The authors should include a privacy analysis of the generated output, as done in [2].

+ The strength of the proposed method is unclear.
Based on the results in Table 9, this approach does not appear to generalize well to open-domain QA datasets.
Additionally, it is not specifically designed for the biomedical domain.
For example, the authors could more carefully design the self-critic prompt to incorporate essential domain knowledge for each biomedical task.
Another concern is the computational efficiency of the proposed method.
While MCTS is effective for complex reasoning tasks, it is known to be computationally costly.
I suggest that the authors include an efficiency analysis to compare the performance gains against the computational cost.


[1] S. Soni and K. Roberts, Evaluation of Dataset Selection for Pre-Training and Fine-Tuning Transformer Language Models for Clinical Question Answering, LREC 2020

[2] X. Zhang et al., Enhancing Small Medical Learners with Privacy-preserving Contextual Prompting, ICLR 2024

[3] S.V. Dibbo et al., SoK: Model Inversion Attack Landscape: Taxonomy, Challenges, and Future Roadmap, CSF 2023

**Questions:**

+ Is $p_{doc}$ a predefined constant? If so, this should be clarified in line 205. Additionally, the pseudocode in the Appendix should be moved to the main article to improve understanding of the expansion phase.

+ Is there a reason why some of the methods listed in Table 2 are not included as baselines?

---

> ### Author Response · Authors · 2024-11-25
> **Response to your review -- part 1**
>
> We thank this reviewer for taking the time to review our work. We have carefully considered each point of feedback and provide our point-by-point responses below. Please don’t hesitate to let us know if any further clarifications are required.
>
> - (P1) Cost efficiency analysis
> - (P2) Additional baselines.
> - (P3) Privacy Analysis.
> - (P4) Prompt engineering.
> - (P5) Clarifications and Presentation
>
> ## **P1 Cost efficiency analysis**
>
> We thank the reviewer's nice suggestion to provide an additional analysis regarding the inference cost of our method. Therefore we conducted an additional comparison of the token costs for each benchmarked method, which is provided in the global response. This new analysis improves the analysis against the baselines relevant to the reader and other reviewers.
>
> ## **P2 Additional baselines.**
>
> We thank the reviewer for the insightful suggestion about including the baseline proposed by S. Soni and K. Roberts [1]. However, this approach is not directly comparable to our work. Their method bypasses the information retrieval (IR) step by directly providing the golden passage—the exact passage containing the necessary information for answering the question—to the LLM. This differs fundamentally from our work, which emphasizes the integration of LLMs with an IR system and seeks to improve robustness to imperfect retrieval.
>
> Unlike the method of S. Soni and K. Roberts [1], our approach does not assume a perfect IR system. Instead, we address the realistic scenario where retrieval outputs may be noisy or incomplete. The goal of our method is to enhance the LLM's ability to provide accurate answers despite such imperfections in the retrieval process. This is a critical advancement, as IR systems in practice often retrieve documents with varying levels of relevance or accuracy.
>
> Additionally, our benchmark already incorporates a recent and related baseline, SELF-RAG [2], which fine-tunes LLMs to better align with IR systems. By including SELF-RAG, we evaluate our method against an approach that integrates LLM fine-tuning and IR optimization, offering a more relevant comparison than [1]. As shown in Table 4 of the manuscript, our proposed RATP framework outperforms SELF-RAG and other baselines in both private and public dataset scenarios, demonstrating its robustness and versatility.
>
> Lastly, we agree with the reviewer that the work by X. Zhang et al. [3] is highly relevant. Its connection to our study is elaborated in the subsequent section.
>
> [1] S. Soni and K. Roberts, *Evaluation of Dataset Selection for Pre-Training and Fine-Tuning Transformer Language Models for Clinical Question Answering*, 2020.
>
> [2] Asai et al., *Self-RAG: Learning to Retrieve, Generate, and Critique through Self-Reflection*, 2023.
>
> [3] X. Zhang et al., *Enhancing Small Medical Learners with Privacy-preserving Contextual Prompting*, 2024.
>
> ## **P3 Privacy Analysis**
>
> We appreciate the reviewer’s suggestion to include the work of X. Zhang et al. [1] in our discussion of related literature. Their study on contextual privacy is indeed relevant, and we agree that it merits inclusion in our related work section. However, as outlined in the limitations section of our paper, our primary focus is on preventing the leakage of sensitive data from the training dataset. This differs from the contextual privacy issues discussed in [1], which involve the protection of data within interactions at inference time. As such, the privacy analysis proposed in [1] is not directly applicable to our current framework, which concentrates on the integrity and privacy of training data during model deployment.
>
> Additionally, while [1] tackles privacy-related challenges effectively, it does not involve the integration of IR systems with LLMs, a central aspect of our work. This distinction makes [1] unsuitable as a baseline for our methodology, which focuses on coupling IR with LLMs to enhance private data handling without retraining models on sensitive data.
>
> Nevertheless, we recognize the potential of combining the approaches discussed in [1] with our framework to address broader privacy concerns. For instance, methods like those proposed in [1] could be adapted to mitigate contextual privacy risks in our thought-generation and scoring components, paving the way for more comprehensive privacy safeguards.
>
> **Update:** The related work section will be expanded to include a discussion of contextual privacy and to highlight the contributions of X. Zhang et al. [1], emphasizing how their work complements but does not directly overlap with our approach.
>
> [1] Enhancing Small Medical Learners with Privacy-preserving Contextual Prompting, 2024.

---

> > ### Author Response · Authors · 2024-11-25
> > **Response to your review -- part 2**
> >
> > ## **P4 Prompt engineering.**
> >
> > We thank the reviewer for their valuable suggestion on incorporating domain knowledge into the self-critic model via prompt engineering. This approach could indeed improve the performance of both the baseline methods and RATP by aligning prompts more closely with the specific task.
> >
> > In our study, we deliberately limited prompt fine-tuning for both the proposed method and the baselines. Heavy reliance on domain-specific knowledge may reduce the adaptability of the approach to varied environments and tasks. This adaptability is a key advantage of RATP, particularly in fields like healthcare, where diverse tasks would require significant prompt tuning to integrate expert knowledge. By maintaining a generalized framework, our method highlights RATP's ability to leverage external knowledge effectively while ensuring robustness across different conditions.
> >
> > Moreover, integrating domain-specific knowledge into prompts could introduce confounding factors, making it unclear whether performance improvements stem from the method itself or the additional knowledge embedded in the prompts. Notably, our method achieves superior performance compared to the baselines even without domain-specific prompt adjustments (see Section 4.2 of the manuscript).
> >
> > Finally, optimizing prompts for each baseline and dataset would require exploring a vast and diverse landscape of existing prompt engineering techniques, which is impractical to implement fairly across the five benchmarked methods, each requiring distinct prompts.
> >
> > **Update:** We will add clarifications in the Appendix to address the integration of domain knowledge via LLM prompts.
> >
> > ## **P5 Clarifications and Presentation**
> >
> > ### Table 2 :
> >
> > The contribution of our paper is also to provide a formalism to think and compare different LLM and Information Retrieval pairing as a sequential decision-making problem.  Table 2 demonstrates how multiple methods align with our proposed framework. However, not all methods are suitable as baselines:
> >
> > - **GOT [1]:** This method requires a handcrafted pattern, which is not apparent for the task at hand.
> > - **RAT [2]:** Its performance is strictly inferior to RATT [3].
> > - **HippoRAG [4]:** This method is infeasible for our public dataset, which is based on a Wikipedia dump. The preprocessing cost for 21 million documents exceeds reasonable limits (estimated at over $6,000, per HippoRAG’s authors).
> >
> > ### P_doc Consistency :
> >
> > We confirm that the `P_doc` value remains constant throughout the experiments.
> >
> > ### Pseudo Code Inclusion :
> >
> > We thank the reviewers for their suggestion to include the pseudo code from Appendix G in the main text. However, due to page constraints, this is not feasible. Instead, we propose to highlight a reference to the pseudo code in the main text to guide readers effectively.
> >
> > [1] Graph of Thoughts: Solving Elaborate Problems with Large Language Models, 2024
> >
> > [2] Rat: Retrieval augmented thoughts elicit context-aware reasoning in long-horizon generation, 2024
> >
> > [3] RATT: AThought Structure for Coherent and Correct LLMReasoning, 2024
> >
> > [4] HippoRAG: Neurobiologically Inspired Long-Term Memory for Large Language Models, 2024
> >
> > **Update:** Clarifications regarding `P_doc` and Table 2 will be added to the revised paper. A reference to the pseudo-code in Appendix G will be prominently highlighted in the main text.
> >
> > ---
> >
> > We hope that we have sufficiently addressed the majority of the reviewers’ concerns and that this may encourage a reconsideration of their scores. We remain available and eager to engage in further discussions.

---

### Official Review · Reviewer_CfV5 · 2024-11-05

**Soundness:** 2
**Presentation:** 3
**Contribution:** 2
**Rating:** 6
**Confidence:** 3

**Summary:**

Large Language Models (LLMs) two major challenges in healthcare such as:
1. Privacy concerns with sensitive medical data
2. Difficulty in reliably accessing and using external knowledge

RATP addresses these by formulating thought generation as a markov decision process where states are reasoning graphs and actions are combinations of thoughts with retrieved documents. The privacy challenge is addressed by keeping sensitive data separate from LLM training, using only frozen inference with retrieved chunks. For retrieval integration, RATP treats documents as potential thoughts in the state space, allowing MCTS to optimize the document selection and reasoning paths, unlike standard RAG which uses fixed retrieval.

The paper shows strong results on benchmarks against their baselines.

**Strengths:**

The paper is well written, it improves upon ToT by unifying document retrieval and thought generation in a single MDP framework, allowing for dynamic graph expansion rather than fixed-tree exploration. The method requires no LLM fine-tuning, preserving privacy and allows easy deployment. The method scales across different LLMs. The papers shows strong improvements against baselines experiments.

**Weaknesses:**

My main concerns with the paper is that it doesn’t compare against strong methods that have been established in the literature such as ReACT [1] or DSP [2] with the only one being ToT. Compared to ToT, how does it compare on LLM calls, total input + output tokens, and time?  The evaluation of retrieval quality is incomplete - there's no analysis of whether MCTS truly finds optimal documents or just compensates for poor retrieval through more reasoning. Does the RAG baseline incorporate Chain of Thought?

[1] Yao et al. ReAct: Synergizing Reasoning and Acting in Language Models
[2] Khattab et al. DEMONSTRATE–SEARCH–PREDICT: Composing retrieval and language models for knowledge-intensive NLP

**Questions:**

1. How does the method compare to ToT in terms of computational costs such as  LLM calls, total input + output tokens, and total time to answer a question?
2. How does the method compare to methods such as ReACT or DSP?
3. The RAG baseline implementation lacks clarity - was chain-of-thought prompting used? How much of RATP's improvement comes from better reasoning versus better retrieval?
4. How does RATP's document selection compare to retrieval metrics (precision@k, recall@k)? Is MCTS actually finding better documents or just processing more of them?

---

> ### Author Response · Authors · 2024-11-25
> **Response to your review -- part 1**
>
> We thank this reviewer for taking the time to review our work. We have carefully considered each point of feedback and provide our point-by-point responses below. Please don’t hesitate to let us know if any further clarifications are required.
>
> - (P1) Efficiency analysis
> - (P2) Additional baselines
> - (P3) Does MCTS find the optimal document?
> - (P4) Prompt engineering
>
> ## **P1 Cost efficiency analysis**
>
> We thank the reviewer's nice suggestion to provide an additional analysis regarding the inference cost of our method. Therefore we conducted an additional comparison of the token costs for each benchmarked method, which is provided in the global response. This new analysis improves the analysis against the baselines relevant to the reader and other reviewers.
>
> ## **P2 Additional baselines**
>
> We appreciate the reviewer's suggestion to compare our framework against additional baselines. To address this, we have included ReAct [1] in our benchmark. The ReAct prompting method enhances reasoning in LLMs by combining reasoning traces with action steps in the same prompt. This approach enables the model to reason through complex problems while simultaneously retrieving or validating necessary information, forming a feedback loop that improves accuracy and coherence. Results for the public dataset and one private dataset (emrQA) due to time constraints are presented below:
>
> | **Dataset** | **LLM** | **RAG** | **Self-RAG** | **RATT** | **ReAct** | **RATP** |
> | --- | --- | --- | --- | --- | --- | --- |
> | Private (emrQA) | 34 (±0.6) | 24 (±0.4) | 35 (±0.5) | 28 (±0.9) | 39 (±1.1) | **71 (±0.5)** |
> | Public (BoolQA) | 66 (±0.8) | 67 (±0.6) | 67 (±0.7) | 71 (±1.1) | 68 (±0.9) | **72 (±0.8)** |
>
> Empirically, ReAct outperforms most baselines, especially for the private dataset, but remains less effective than RATP. This performance gap arises from differences in their reasoning processes:
>
> The performance gap compared to RATP can be attributed to differences in their thought processes. ReAct concatenates all actions (retrieval steps or thought generation) into a single sequence, while RATP generates each thought independently based on a small, carefully selected set of thoughts or documents. This distinction has two key impacts:
>
> 1. ReAct retains all thoughts and documents, including irrelevant or noisy ones, to generate new actions. In contrast, RATP uses a scoring model to filter and prioritize relevant inputs, reducing confusion.
> 2. The concatenation of all prior actions in ReAct often exceeds the context size window, limiting the thought process. RATP avoids this issue by generating each thought independently within the context window.
>
> These differences explain RATP's advantage in handling tasks that rely on imperfect retrieval. Moreover, this sensitivity to unsuccessful retrieval has been noted by ReAct's author in their paper.
>
> [1] ReAct: Synergizing Reasoning and Acting in Language Models, 2023
>
> **Update:** the benchmark section of the revised paper will be updated to include ReAct.

---

> > ### Author Response · Authors · 2024-11-25
> > **Response to your review -- part 2**
> >
> > ## **P3 Impact of Retrieval vs. Reasoning**
> >
> > We agree with the reviewer that understanding the contributions of both retrieval and reasoning to the method's performance is an important question. However, we first emphasize that while RATP can influence retrieval by formatting queries, its primary goal is not to improve document retrieval directly. Instead, the retrieval step is consistently performed by the same model (COntriever) across all experiments. RATP focuses on optimizing the pairing of a large language model (LLM) with a retrieval system, aiming to make the best use of imperfect retrieval results, particularly by enhancing robustness against irrelevant documents.
> >
> > Additionally, our experimental setup does not facilitate standard retrieval metrics, as we do not rely on a predefined collection of golden paragraphs. Instead, documents are retrieved directly from real-world databases (e.g., Wikipedia dumps or EMR passages), making it challenging to reliably flag relevant documents. Nevertheless, to explore the impact of retrieval on performance, we include a comparison in Appendix 10. This comparison evaluates RATP's performance with and without information retrieval (IR) using the F1 and Rouge-L metrics, which assess the similarity between the generated thoughts and the golden paragraphs from the BoolQA dataset (not used during training). We also include the Oracle metric, representing the oracle score on the generated thoughts. As shown in the table below, the results indicate that IR improves the quality of generated thoughts across all three metrics, reducing hallucinations.
> >
> > **Table 1:** Assessment of thought quality with and without IR. The frozen LLM used is Mixtral8x7B, and the questions are from the BoolQA dataset.
> >
> > | **Metric** | **MCTS Oracle w/ IR** | **MCTS Oracle w/o IR** |
> > | --- | --- | --- |
> > | **F1** | 0.134 | 0.080 |
> > | **Rouge-L** | 0.082 | 0.052 |
> > | **Oracle** | 0.317 | 0.255 |
> >
> > Characterizing the impact of reasoning is similarly nontrivial. However, we can demonstrate that improved reasoning enhances performance. In RATP, reasoning is significantly influenced by the scoring system, which provides feedback, selects thoughts for further development, and determines when to terminate the algorithm. To illustrate this, Appendix E compares RATP's performance using increasingly proficient scoring systems while keeping the same frozen LLM for thought and answer generation. The results, presented below, show a clear correlation between better scoring models and improved RATP performance.
> >
> > **Table 2:** Comparison of RATP (MCTS self-critic) performance on the BoolQA dataset using different LLMs as critic models. The thought generation LLM is consistently Llama 70B.
> >
> > | **Self-critic Model** | **Accuracy** |
> > | --- | --- |
> > | **Llama-2 70B** | 70 |
> > | **GPT3.5-turbo** | 73 |
> > | **GPT4** | 81 |
> > | **Oracle** | 83 |
> >
> > In conclusion, RATP's improved performance arises from both the quality of information retrieval and the reasoning capabilities of the scoring system.
> >
> > **Update:** The insights from these results will be highlighted in the main text in the revised manuscript.

---

> > > ### Author Response · Authors · 2024-11-25
> > > **Response to your review -- part 3**
> > >
> > > ## **P4 RAG baseline and prompt engineering**
> > >
> > > **Implementation of the RAG Baseline.** To clarify the implementation of the RAG baseline, we retrieve the most relevant document from the database using an Information Retrieval model (Contriever), with the question serving as the query. This retrieved document is combined with the question using a predefined prompt template. Examples of such templates are shown in Figure 6 (emrQA) and Figure 12 (boolQA) in the Appendix.
> > >
> > > **Chain-of-Thought Prompting.** Thank you for pointing this out. To clarify, if "Chain-of-Thought" (CoT) prompting refers to the technique of providing a series of expert demonstrations to guide the LLM before presenting a question (as described in [1]), we do not use this method in our approach or for any of the baseline models. While prompt engineering techniques such as CoT are widely applicable, fine-tuning each prompt for every dataset is infeasible given the large variety of possible configurations (5 benchmarked methods, each with different prompts).
> > >
> > > If "Chain-of-Thought" instead refers to a step-by-step reasoning process, with or without self-consistency, this is indeed a specific case of the Tree-of-Thought (ToT) approach. In our work, ToT with Information Retrieval is benchmarked as part of the ablation studies (see Tables 3 and 9). Specifically, CoT corresponds to a tree structure with a size and breadth limit of 1, which is included in the more general ToT framework. The ToT method has been shown to achieve a strictly superior performance compared to CoT [2].
> > >
> > > **Update:** We will incorporate additional details in the experimental section of the appendix to further elucidate the implementation of the RAG baseline. Furthermore, we will clarify the usage and comparison of "Chain-of-Thought" prompting in the related work section.
> > >
> > > [1] Chain-of-Thought Prompting Elicits Reasoning in Large Language Models, 2023
> > >
> > > [2] Tree of Thoughts: Deliberate Problem Solving with Large Language Models, 2023
> > >
> > > ---
> > >
> > > We hope that we have sufficiently addressed the majority of the reviewers’ concerns and that this may encourage a reconsideration of their scores. We remain available and eager to engage in further discussions.

---

> > > > ### Comment · Reviewer_CfV5 · 2024-11-27
> > > >
> > > > Thank you for answering my questions. I would still be interested in seeing comparisions against other stronger baselines but doing this in the limited rebuttal period is difficult and you have shown some better results than ReACT so I have updated my score.

---

### Author Response · Authors · 2024-11-25
**Global Response**

We sincerely thank the reviewers for their valuable and constructive feedback. We are encouraged by the reviewers’ recognition of the novelty and effectiveness of our proposed Retrieval-Augmented Thought Process (RATP) framework in addressing key challenges related to private data handling and reasoning in healthcare applications.

$\color{red} CfV5$: “The method requires no LLM fine-tuning, preserving privacy and allows easy deployment. The papers show strong improvements against baseline experiments. ”

$\color{blue} 7Jne$: “The RATP addresses a particular type of privacy concerns which are highly relevant in the healthcare domain: the use of sensitive data during training. By avoiding this, the approach enhances the applicability of LLMs in handling sensitive healthcare information safely and efficiently.”

$\color{magenta} XAJF$: “This paper takes the privacy question in healthcare into consideration, which is very critical in the healthcare domain, and proposes corresponding methods to enhances the interaction between LLMs and private data.”

$\color{magenta} vqsA$: “This method is extremely useful in the scenarios where private data is needed to answer a question.”

---

Reviewers also had questions about the cost efficiency of the method. To address this, we provide a detailed analysis of token costs for each method below and respond to individual concerns in the respective rebuttals. We believe this point, in combination with the reviewer’s addressed concerns, significantly strengthens our paper. Thank you for your valuable feedback.

## P1 Cost efficiency analysis

We thank the reviewers for suggesting an analysis of the cost efficiency of the methods. We agree that presenting inference costs in terms of tokens provides valuable guidance for practitioners to assess the practicality of RATP.

In the table below, we provide a benchmark of the average token cost (± standard error) per question. Due to time constraints and limited computational resources, we reran all methods on the public setting dataset and one of the two private datasets. We observe that RATP, which utilizes Monte Carlo Tree Search (MCTS) incurs higher token costs than most baseline methods on average with the exception of Tree of Thought (ToT) with Information Retrieval (IR). Notably, the cost increase varies significantly by task, for example, the cost increase for the open-domain public setting is relatively modest compared to the private closed-domain setting, where costs rise by a factor of 10. This aligns with the performance gaps observed, as RATP’s advantages over baselines are more pronounced in the private setting. This indicates that RATP is particularly beneficial for private closed-domain tasks.

Finally, RATP is more cost-efficient than ToT with IR, underscoring the advantages of using MCTS over tree search.

| **Dataset** |  | **LLM** | **RAG** | **RATT** | **Self-RAG** | **ToT w/ IR** | **RATP** |
| --- | --- | --- | --- | --- | --- | --- | --- |
| emrQA | Input tokens | 70 (± 0) | 186 (± 9) | 4914 (± 130) | 2199 (± 27) | 12511 (± 71) | 9416 (± 82) |
|  | Output tokens | 4 (± 0) | 4 (± 0) | 1048 (± 16) | 530 (± 14) | 1449 (± 20) | 1180 (± 24) |
| BoolQA | Input tokens | 66 (± 0) | 226 (± 2) | 5262 (± 150) | 1929 (± 12) | 5481 (± 30) | 662 (± 60) |
|  | Output tokens | 1 (± 0) | 1 (± 0) | 1330 (± 31) | 395 (± 24) | 935 (± 9) | 76 (± 11) |

We also note that this analysis focuses solely on inference costs and does not account for training costs, which are particularly relevant for methods like Self-RAG. Fine-tuning large language models can make adapting to new tasks prohibitively expensive, especially in the healthcare domain, where the diversity of tasks (described in Appendix A of the manuscript) necessitates constant retraining.

**Update:** the cost efficiency analysis discussed above will be integrated into a new Appendix in the revised paper.

---

### Meta-Review · Area_Chair_yUbJ · 2024-12-21

**Metareview:**

The paper introduces Retrieval-Augmented Thought Process (RATP), leveraging Monte Carlo Tree Search to enhance privacy-preserving reasoning in healthcare, achieving superior accuracy and robustness against imperfect retrieval in question-answering tasks.

Strengths
- RATP combines retrieval-augmented reasoning and Monte Carlo Tree Search to optimize decision-making in LLMs while ensuring privacy, addressing critical healthcare problems.
- RATP achieves up to 35% accuracy improvements on private datasets, demonstrating its effectiveness compared to baseline methods like RAG and Tree-of-Thought.

Weaknesses
- The paper relies  heavily on existing methods (e.g., retrieval-augmented generation, multi-step reasoning) without introducing fundamentally new ideas, significantly reducing the originality of its contribution.
- The lack of comparisons with well-established methods like Chain-of-Thought and other relevant baselines (e.g., ReAct, ClinicalBERT) weakens the evaluation of the proposed framework's benefits
- RATP’s multi-step reasoning with MCTS significantly increases inference costs, raising doubts about its practicality for real-time or resource-constrained applications

**Additional Comments On Reviewer Discussion:**

During the rebuttal, authors added one of the baselines (ReAct) requested by the reviewers, but rejected to include CoT, claiming that it is a special case of Tree-of-Thought. As for increased cost due to MCTS, authors provided some scenarios where RATP could be potentially cost-effective. Another reviewer concern regarding contextual privacy was not addressed, while authors have clarified it in the limitations section. But overall, the reviewers were not convinced of the originality of the paper’s methodological contribution.

---

### Decision · Program_Chairs · 2025-01-22

Reject